# Variational Inference via Resolution of Singularities

## Abstract

Predicated on the premise that neural networks are best viewed as singular statistical models, we set out to propose a new variational approximation for Bayesian neural networks. The approximation relies on a central result from singular learning theory according to which the posterior distribution over the parameters of a singular model, following an algebraic-geometrical transformation known as a desingularization map, is asymptotically a mixture of standard forms. From here we proceed to demonstrate that a generalized gamma mean-field variational family, following desingularization, can recover the leading order term of the model evidence. Affine coupling layers are employed to learn the unknown desingularization map, effectively rendering the proposed methodology a normalizing flow with the generalized gamma as the source distribution.

## 1 Introduction

Singular statistical models are ubiquitous in modern machine learning. In contrast to their regular counterpart, singular models need not be identifiable nor possess a positive-definite Fisher information matrix. These departures from classic regularity conditions have considerable implications for both theory and practice. Since neural networks are (strictly) singular models, accounting for such differences may be critical to some of the most pressing challenges in deep learning theory including model selection and the generalization puzzle (Murfet et al., 2020).

In this work, singular learning theory (Watanabe, 2009) is brought to bear on the challenge of performing inference in Bayesian neural networks (MacKay, 1992; Neal, 1996). In particular, singular learning theory helps us understand the large-sample properties of the posterior distribution over neural network weights $w \in \mathbb{R}^d$. By (Watanabe, 2018, Chapter 6), there exists a so-called **desingularization map** (also known as a **resolution map**), $g : \mathbb{R}^d \to \mathbb{R}^d, g(\xi) = w$, such that when $n$ is large, the posterior distribution in a local weight set is proportional to

$$\exp(-n\xi_1^{2k_1}\xi_2^{2k_2}\cdots\xi_d^{2k_d})|\xi_1^{h_1}\cdots\xi_d^{h_d}|b(\xi). \tag{1}$$

The $k$'s and $h$'s are natural numbers, and $b(\cdot)$ is a real-valued $C^\infty$ function.

We say the posterior distribution has been put into *standard form* if coordinates $\xi$ have been found that allow the posterior to be locally written as in equation 1. Remarkably, it is possible under very general conditions to write the posterior distribution over the parameters of a singular model as just such *a mixture of standard forms*. Based on equation 1, we proceed to demonstrate that a certain mean-field variational family, following desingularization, can recover the leading order term of the log (normalized) evidence, up to constant terms that do not depend on sample size $n$ or dimension $d$.

Let $(x, y)$ denote the input-target pair modeled jointly as $p(x, y|w) = p(y|x, w)p(x)$. Let us assume the parameter space $W$ is a compact set in $\mathbb{R}^d$ and $p_0(x, y) = p_0(y|x)p(x)$ is the true data-generating mechanism. Throughout we suppose there exists $w_0 \in W$ such that $p_0(y|x) = p(y|x, w_0)$. In the parlance of singular learning theory, this condition is known as **realizability**. Let $\varphi(w)$ be a compactly-supported prior. We shall refer to $(p(\cdot, \cdot), p_0(\cdot, \cdot), \varphi(\cdot))$ as a **model-truth-prior triplet**.

Define $K(w)$ to be the Kullback-Leibler divergence between the truth and the model, as a function of the model parameters $w$:

$$K(w) = \mathrm{KL}(p_0(x, y)||p(x, y|w)) = E_{p_0} \log \frac{p_0(y|x)}{p(y|x, w)}.$$

Following Watanabe (2009), we say a model is **regular** if it is 1) identifiable i.e., $\{w : K(w) = 0\}$ is a singleton, and 2) its Fisher information matrix $I(w)$ is positive definite for arbitrary $w \in W$. We call a model **strictly singular** if it is not regular. The term singular will refer to either regular or strictly singular models.

Henceforth let $p(y|x, w)$ be a neural network model. We wish to approximate the intractable posterior distribution over neural network weights, $p(w|\mathcal{D}_n) = \frac{\prod_{i=1}^n p(y_i|x_i, w)\varphi(w)}{Z(n)}$, where $\mathcal{D}_n = \{(x_i, y_i)\}_{i=1}^n$ is a dataset of $n$ input-output pairs. The normalizing constant, $Z(n) = \int \prod_{i=1}^n p(y_i|x_i, w)\varphi(w)\, dw$, is variously known as the (model) **evidence** and the **marginal likelihood**.

The asymptotic expansion of $\bar{Z}_K(n)$, a variant of $Z(n)$ defined in equation 5, will prove crucial to justifying the proposed variational approximation. For strictly singular models, quantities such as $Z(n)$ and $\bar{Z}_K(n)$ manifest as a singular integral, i.e., an integral of the form $\int_W e^{-nf(w)}\varphi(w)\, dw$ where $W \subset \mathbb{R}^d$ is a compact semi-analytic subset, and $f$ and $\varphi$ are real analytic functions. The behavior of a singular integral critically depends on the zeros of $f$.

## 2 VARIATIONAL INFERENCE FOR SINGULAR MODELS

Variational inference is an approximate inference technique in which a family of densities $\mathcal{Q}$, often called the variational family, is first posited and a member of the variational family, some $q^* \in \mathcal{Q}$, is then found via optimization according to some criterion that measures closeness to the desired target density. To begin, let us write the posterior distribution $p(w|\mathcal{D}_n)$ in two alternate forms:

$$L_n(w) = -\frac{1}{n}\sum_{i=1}^n \log p(y_i|x_i, w) \qquad\qquad K_n(w) = \frac{1}{n}\sum_{i=1}^n \log \frac{p_0(y_i|x_i)}{p(y_i|x_i, w)}$$

$$p(w|\mathcal{D}_n) \propto e^{-nL_n(w)}\varphi(w) \qquad\qquad p(w|\mathcal{D}_n) \propto e^{-nK_n(w)}\varphi(w)$$

$$Z(n) = \int_W e^{-nL_n(w)}\varphi(w)\, dw \qquad\qquad \bar{Z}(n) = \int_W e^{-nK_n(w)}\varphi(w).$$

On the left, the posterior is written in terms of the average negative log likelihood $L_n(w)$ and on the right in terms of the average log likelihood ratio $K_n(w)$, which is the empirical counterpart to $K(w)$. Since $\bar{Z}(n) = Z(n)/\prod_{i=1}^n p_0(y_i|x_i)$, we refer to $\bar{Z}(n)$ as the **normalized evidence**.

Now, consider some general change of variables $g : \mathbb{R}^d \to \mathbb{R}^d, g(\xi) = w$. In the new coordinate $\xi$, the KL divergence between a variational distribution $q(\xi)$ and the desired posterior target,

$$p(\xi|\mathcal{D}_n) \propto e^{-nK_n(g(\xi))}\varphi(g(\xi))|g'(\xi)|,$$

is given by

$$\mathrm{KL}(q(\xi)||p(\xi|\mathcal{D}_n)) = E_q nK_n(g(\xi)) + \mathrm{KL}(q(\xi)||\varphi(g(\xi))|g'(\xi)|) + \log\bar{Z}(n).$$

As long as the support of $q$ is contained in the support of the posterior, we have $\mathrm{KL}(q(\xi)||p(\xi|\mathcal{D}_n)) \geq 0$ and hence the following bound:

$$\Psi(q, g) := -E_q nK_n(g(\xi)) - \mathrm{KL}(q(\xi)||\varphi(g(\xi))|g'(\xi)|) \leq \log\bar{Z}(n). \tag{2}$$

Equality in equation 2 is achieved if and only if $q(\xi) = p(\xi|\mathcal{D}_n)$. It is easy to recognize that maximizing $\Psi(q, g)$ is equivalent to maximizing the so-called evidence lower bound,

$$\mathrm{ELBO}(q, g) := E_q\left[\sum_{i=1}^n \log p(y_i|x_i, g(\xi)) - \log q(\xi) + \log(\varphi(g(\xi))|g'(\xi)|)\right], \tag{3}$$

since $\Psi(q, g) = \mathrm{ELBO}(q, g) + nS_n$ where $S_n = -\frac{1}{n}\sum_{i=1}^n \log p_0(y_i|x_i)$ is the empirical entropy. Indeed, just as $\Psi(q, g)$ is a lower bound on the log (normalized) evidence $\log\bar{Z}(n)$, so too is $\mathrm{ELBO}(q, g)$ a lower bound on the log (unnormalized) evidence $\log Z(n)$.

To facilitate theoretical analysis, we will work with the deterministic counterparts to $\Psi(q, g)$ and $\bar{Z}(n)$, respectively given by

$$\Psi_K(q, g) := -E_q nK(g(\xi)) - \mathrm{KL}(q(\xi)||\varphi(g(\xi))|g'(\xi)|), \tag{4}$$

and

$$\bar{Z}_K(n) := \int_W e^{-nK(w)} \varphi(w) \, dw. \tag{5}$$

It is plain to see that just as $\Psi(q, g) \leq \log \bar{Z}(n)$, we have $\Psi_K(q, g) \leq \log \bar{Z}_K(n)$. **Our theoretical results address the possibility that** $\sup_{q \in \mathcal{Q}} \Psi_K(q, g) \approx \log \bar{Z}_K(n)$ **for some variational family** $\mathcal{Q}$ **and change-of-variables** $g$.

By (Watanabe, 2009, Theorem 6.7), for the leading order term in the asymptotic expansion of $\bar{Z}_K(n)$, we have

$$\bar{Z}_K(n) \approx C n^{-\lambda} (\log n)^{m-1}. \tag{6}$$

The rational number $\lambda \in [0, d/2]$ is an important quantity in singular learning theory known as the real log canonical threshold (RLCT) and the integer $m \geq 1$ is its associated multiplicity. These two quantities, to be defined in equation 11, are uniquely associated to a model-truth-prior triplet.

Equipped with equation 6, (Bhattacharya et al., 2020, Theorem 3.1) established that when $g$ is a resolution map, the mean-field variational family $\mathcal{Q}_{(0,1]}$ of equation 12 can achieve

$$\sup_{q \in \mathcal{Q}_{(0,1]}} \Psi_K(q, g) \geq -\lambda \log n - \text{ constant independent of n.} \tag{7}$$

(In Appendix C, we prove this result under more general conditions than assumed in Theorem 3.1 of Bhattacharya et al. (2020).) Unfortunately, the bound in equation 7 provides little reassurance that $\mathcal{Q}_{(0,1]}$ is a desirable variational family.

**Contribution**  Following Lin (2011), let us call $C$ in equation 6 the **leading coefficient** of $\bar{Z}_K(n)$. We go beyond the analysis in Bhattacharya et al. (2020) by taking into account those terms in the leading coefficient that depend on the dimension $d$, call it $C(d)$. Our main result, Theorem 5.1, shows that if $g$ is a resolution map, then the same variational family $\mathcal{Q}_{(0,1]}$ of equation 12 can achieve

$$\sup_{q \in \mathcal{Q}_{(0,1]}} \Psi_K(q, g) = -\lambda \log n + \log C(d). \tag{8}$$

Next, rather than presuming the resolution map theoretically tractable as in Bhattacharya et al. (2020), we employ a normalizing flow to learn the unknown resolution map $g$ at the same time as learning the variational parameters in $q$. We are aided by the fact that a resolution map can attain the optimal value of $\Psi_K(q, g)$ and therefore justifies learning the resolution map via optimization of $\Psi(q, g)$. Finally let us note that although the result in equation 8 is stronger than that in equation 7, it does come at the cost of additional assumptions as we will discuss at the end of Section 5.

**Remark.** *When the model is regular, we need not bother with singular learning theory and may obtain* $\bar{Z}_K(n) \approx \varphi(w_0) \sqrt{\frac{(2\pi)^d}{\det H(w_0)}} n^{-d/2}$ *via the Laplace approximation. The Laplace approximation, however, is egregiously inappropriate for singular models, in particular neural network models. Since $\lambda = d/2$ and $m = 1$ in regular models, equation 6 is a true generalization of the Laplace approximation, holding for both regular and strictly singular models.*

## 3  SINGULAR LEARNING THEORY

That the posterior distribution in singular models can be written, under quite general conditions, as a mixture of standard forms is predicated on the *monomialization* of $K(w)$. The following theorem from Watanabe (2009), adapted for notational consistency, gives precise conditions for the existence of the resolution map, an algebraic geometrical transformation such that $K(w)$ can be written as a monomial. The result is itself based on Hironaka's resolution of singularities, a celebrated result in modern algebraic geometry. To prepare, let $W_\epsilon = \{w \in W : K(w) \leq \epsilon\}$ for some small positive constant $\epsilon$ and $W_\epsilon^{(R)}$ be some real open set such that $W_\epsilon \subset W_\epsilon^{(R)}$. The theorem below will make use of the multi-index notation: for a given $w = (w_1, \ldots, w_d) \in \mathbb{R}^d$, define $w^{\boldsymbol{k}} := w_1^{k_1} \cdots w_d^{k_d}$ where the multi-index $\boldsymbol{k} = (k_1, \ldots, k_d)$ with each $k_j$ a nonnegative integer.

**Theorem 3.1** (Theorem 6.5 of Watanabe (2009)). *Suppose the model-truth-prior triplet $(p, p_0, \varphi)$ satisfies Fundamental Conditions I and II with $s = 2$ in Watanabe (2009). We can find a real analytic manifold $M^{(R)}$ and a proper and real analytic map $g : M^{(R)} \to W_\epsilon^{(R)}$ such that*

1. $M = g^{-1}(W_\epsilon)$ is covered by a finite set $M = \cup_\alpha M_\alpha$ where $M_\alpha = [0, b]^d$.

2. In each $M_\alpha$,

$$K(g(\xi)) = \xi^{2\boldsymbol{k}} = \xi_1^{2k_1} \cdots \xi_d^{2k_d}, \tag{9}$$

   where $k_j \in \mathbb{N}$ such that not all $k_j$ are zero.

3. There exists $C^\infty$ function $b(\xi)$ such that

$$\varphi(g(\xi))|g'(\xi)| = \xi^{\boldsymbol{h}} b(\xi) = \xi_1^{h_1} \cdots \xi_d^{h_d} b(\xi), \tag{10}$$

   where $h_j \in \mathbb{N}$, $|g'(\xi)|$ is the absolute value of the determinant of the Jacobian and $b(\xi) > c > 0$ for $\xi \in [0, b]^d$.

If we "plug in" equation 9 and equation 10 into the transformed posterior $p(\xi|\mathcal{D}_n)$, we obtain the first display of the paper, equation 1. Theorem 6.5 of Watanabe (2009) holds for regular statistical models as well, e.g., by the transform $w = g(\xi) = w_0 + I(w_0)^{1/2}\xi$, we can put a regular model-truth-prior triplet into the standard form. It is worth noting that neither the resolution map $g$ nor the multi-indices $\boldsymbol{k}$ and $\boldsymbol{h}$ are unique for a given triplet $(p, p_0, \varphi)$.

A crucial quantity that appears in singular learning theory is the real log canonical threshold (RLCT). Let $\{M_\alpha : \alpha\}$ be as in Theorem 3.1 and $\lambda_j = \frac{h_j + 1}{2k_j}, j = 1, \ldots, d$ where $h_j$ and $k_j$ are the entries of the multi-indices $\boldsymbol{h}$ and $\boldsymbol{k}$ in a local coordinate $M_\alpha$. (For brevity, the dependence on $\alpha$ has been suppressed.) When $k_j = 0$, $\lambda_j$ is taken to be infinity. Uniquely associated to a triplet $(p, p_0, \varphi)$ are its real log canonical threshold (RLCT) and its multiplicity defined, respectively, as

$$\lambda = \min_\alpha \min_{j \in 1, \ldots, d} \lambda_j, \quad m = \max_\alpha \#\{j : \lambda_j = \lambda\}. \tag{11}$$

Let $\{\alpha^*\}$ be the set of those local coordinates in which both the $\min$ and $\max$ in equation 11 are attained. Watanabe (2000) calls such a set the **essential coordinates**.

Assuming the prior is proper, the RLCT of a model-truth-prior triplet is *at most* $d/2$ (Watanabe, 2009, Theorem 7.2). When the model is regular, the RLCT is *exactly equal* to $d/2$ and the multiplicity $m = 1$ (Watanabe, 2009, Remark 1.15). Murfet et al. (2020) argues that (twice) the RLCT might be a most natural count of parameters in singular models. In Appendix A, we recall a simple toy example where the resolution map, the RLCT, and the multiplicity can be calculated explicitly.

**Remark.** *In Section 6, we will learn the resolution map using affine coupling layers. However as indicated by Theorem 3.1, the resolution map $g(\xi)$ as well as the multi-indices $\boldsymbol{k}$ and $\boldsymbol{h}$ in fact depend on $\alpha$. Despite this, it is unclear if learning multiple resolution maps $g_\alpha$ would have any practical advantages since the RLCT is determined entirely by the essential local coordinates.*

## 4 RELATED WORK

Bayesian learning in neural networks well precede the advent of modern deep learning MacKay (1992); Neal (1996). The resurgence of interest in Bayesian learning for deep neural networks, sometimes called Bayesian deep learning Wilson & Izmailov (2020), has been prompted by concerns of overconfidence and miscalibration. Since exact inference for Bayesian neural networks is intractable, all methods proceed by approximate inference. A major class of approximate inference techniques is based on scaling classic MCMC to modern settings of large datasets and deep neural networks. Some scalable variants of MCMC suitable for deep neural networks include Welling & Teh (2011); Chen et al. (2014); Zhang et al. (2019). Another major approximate inference technique for Bayesian neural network is represented by variational inference which learns the target posterior via optimization. The various flavors of variational inference can be commonly characterized by two ingredients: 1) an approximating family, e.g., a class of distributions over the neural network weights and 2) a criterion for measuring closeness of two distributions. The most commonly employed approximating family is undoubtedly the mean-field family of fully factorized Gaussian distributions. Many variational inference techniques share this approximating family even if they use different criterion to measure closeness to the target (Graves, 2011; Blundell et al., 2015; Hernandez-Lobato et al., 2016; Li & Turner, 2016; Khan et al., 2018; Sun et al., 2019). The limitations of the mean-field Gaussian approximating family are well known however (MacKay, 1992). The desire to move beyond

mean-field Gaussian has motivated many recent methods to make use of more realistic covariance structures (Zhang et al., 2018) or more expressive approximating families, e.g., via normalizing flows (Louizos & Welling, 2017). Monte Carlo dropout (Gal & Ghahramani, 2016) is another popular approximate inference technique for Bayesian neural networks which, despite first appearances, can in fact be viewed as variational inference. Another strain of work is based on the idea of using stochastic gradient descent as a sampler for the underlying posterior distribution of interest. Works in this spirit include Mandt et al. (2018), Izmailov et al. (2018) and Maddox et al. (2019). Finally there are various approximate inference techniques for Bayesian neural networks that are not easily classifiable according to the distinctions above, e.g., temperature scaling (Guo et al., 2017) and deep ensembles (Lakshminarayanan et al., 2017).

## 5 THE GENERALIZED GAMMA MEAN-FIELD APPROXIMATION

Consider the mean-field variational family proposed in Bhattacharya et al. (2020)

$$\mathcal{Q}_{(0,1]} = \{q_{\boldsymbol{\lambda},\boldsymbol{k},\boldsymbol{\beta}} = \prod_{j=1}^{d} q_{\lambda_j,k_j,\beta_j}(\xi_j) : \boldsymbol{\lambda} = (\lambda_1,\ldots,\lambda_d) \in \mathbb{R}^d_{>0}, \boldsymbol{k} = (k_1,\ldots,k_d) \in \mathbb{R}^d_{>0}, \boldsymbol{\beta} = (\beta_1,\ldots,\beta_d) \in (0,\infty)^d\}$$

$$q_j(\xi_j) := q_{\lambda_j,k_j,\beta_j}(\xi_j) \propto \xi_j^{2k_j\lambda_j-1} \exp(-\beta_j\xi_j^{2k_j}) 1_{(0,1]}(\xi_j), \tag{12}$$

where each univariate density $q_j$ supported on $(0,1]$ is the density of a truncated generalized gamma random variable. Let $j^* \in \{1,\ldots,d\}$ be the dimension that attains $\tilde{\lambda}_{j^*} = \tilde{\lambda}$ where $\tilde{\lambda}$ is the RLCT of the underlying (unknown) model-truth-prior triplet. It was established in Bhattacharya et al. (2020) that if $g$ is a resolution map, then $\mathcal{Q}_{(0,1]}$ satisfies equation 7 in particular by setting $\boldsymbol{\lambda}$ and $\boldsymbol{k}$ to their respective true values, and $\beta_{j^*} = n$ and all other $\beta_j = 1$.

In Theorem 5.1 below we will instead establish equation 8. In the proof we show this can be accomplished by setting $\boldsymbol{\lambda}$ and $\boldsymbol{k}$ to their true values as in Bhattacharya et al. (2020) though the optimal values of $\boldsymbol{\beta}$ will be different. Finally, although we assume below that $m = 1$, this is not necessary; as long as $m \ll d$, we can set $\beta_j = n^{1/m}$ for all $j$ such that $\tilde{\lambda}_j = \tilde{\lambda}$.

**Theorem 5.1.** *Suppose the model-truth-prior triplet is such that Theorem 3.1 holds with $[0,b] = [0,1]$, $K(g(\xi)) = \xi^{2\tilde{\boldsymbol{k}}}$ and $\varphi(g(\xi))|g'(\xi)| = \xi^{\tilde{\boldsymbol{h}}}$. Let $\tilde{\lambda}$ denote the RLCT of the triplet and assume the multiplicity $m = 1$. Then $\sup_{q \in \mathcal{Q}_{(0,1]}} \Psi_K(q,g) = -\tilde{\lambda}\log n + \log C(d)$ where $C(d)$ is the term in the leading coefficient $C$ that depends on $d$.*

*Proof.* If $\xi \in [0,1]^d$ and $\varphi(g(\xi))|g'(\xi)| \propto \xi^{\tilde{\boldsymbol{h}}}$, then necessarily $b(\xi) = \prod_{j=1}^{d} 1/(\tilde{h}_j + 1)$. Let $\tilde{\lambda}_j = (\tilde{h}_j + 1)/(2\tilde{k}_j)$. Applying Corollary 5.9 in Lin (2011), we obtain

$$\bar{Z}_K(n) = \int_{[0,1]^d} \exp(-n\xi^{2\tilde{\boldsymbol{k}}})\xi^{\tilde{\boldsymbol{h}}} \, d\xi \approx Cn^{-\tilde{\lambda}}(\log n)^{m-1} = Cn^{-\tilde{\lambda}} \tag{13}$$

where

$$C = \frac{\Gamma(\tilde{\lambda})}{(m-1)! \prod_{j=1}^{d}(2\tilde{k}_j) \prod_{j=m+1}^{d}(2\tilde{k}_j)(\tilde{\lambda}_j - \tilde{\lambda})}. \tag{14}$$

If we denote by $C(d)$ the terms in the leading coefficient that depend on $d$, then we have

$$\log C(d) = -\sum_{j=1}^{d} \log(2\tilde{k}_j) - \sum_{j=m+1}^{d} \log(2\tilde{k}_j) - \sum_{j=m+1}^{d} \log \tilde{\lambda}_j - \sum_{j=m+1}^{d} \log(1 - \tilde{\lambda}/\tilde{\lambda}_j).$$

Returning to our variational distribution, if the $k_j$'s are well specified and we additionally set $\beta_{j^*} = n$ where $j^*$ is the dimension that attains $\tilde{\lambda}_{j^*} = \tilde{\lambda}$, we then have $E_q nK(g(\xi)) = \lambda_{j^*} \prod_{j\neq j^*} G(\lambda_j, \beta_j)$,

where $G(\lambda, \beta)$ is as in B.1. Next, we have

$$\mathrm{KL}(q(\xi)||\varphi(g(\xi))|g'(\xi)|) = \mathrm{KL}\Big(q \,\Big\|\, \xi^{\tilde{\boldsymbol{h}}} b(\xi)\Big) = \sum_{j=1}^{d} \mathrm{KL}\Big(q_j \,\Big\|\, \xi_j^{\tilde{h}_j}/(\tilde{h}_j + 1)\Big)$$

$$= \sum_{j=1}^{d} E_{q_j} \log q_j - \tilde{h}_j E_{q_j} \log \xi_j + \log(\tilde{h}_j + 1).$$

If in addition the $\lambda_j$'s are all well-specified, then

$$\mathrm{KL}(q||\varphi(g(\xi))|g'(\xi)|) = \sum_{j=1}^{d} \left[ -\beta_j G(\tilde{\lambda}_j, \beta_j) - \log B(\tilde{k}_j, \tilde{h}_j, \beta_j) + \log(2\tilde{k}_j) + \log \tilde{\lambda}_j \right],$$

where $B(k, h, \beta)$ is as in B.1. Now let us make use of the fact that $\log B(k, h, \beta) \asymp -\lambda \log \beta$ and $G(\lambda, \beta) \asymp \lambda/\beta$ for large $\beta$. Since $\beta_{j^*} = n$ is large, we get

$$\Psi_K(q, g) = -\tilde{\lambda} \prod_{j \neq j^*} G(\tilde{\lambda}_j, \beta_j) + \sum_{j=1}^{d} \left[ \beta_j G(\tilde{\lambda}_j, \beta_j) + \log B(\tilde{k}_j, \tilde{h}_j, \beta_j) - \log(2\tilde{k}_j) - \log \tilde{\lambda}_j \right]$$

$$\asymp -\tilde{\lambda} \log n + \tilde{\lambda}(1 - \prod_{j=m+1}^{d} G(\tilde{\lambda}_j, \beta_j)) + \sum_{j=m+1}^{d} [\beta_j G(\tilde{\lambda}_j, \beta_j) + \log B(\tilde{k}_j, \tilde{h}_j, \beta_j)] - \sum_{j=1}^{d} \log(2\tilde{k}_j) - \sum_{j=1}^{d} \log(\tilde{\lambda}_j).$$

If $\beta_j$ are sufficiently large for $j \neq j^*$, we get

$$\Psi_K(q, g) \asymp -\tilde{\lambda} \log n - \sum_{j=1}^{d} \log(2\tilde{k}_j) - \sum_{j=1}^{d} \log(\tilde{\lambda}_j) + \sum_{j=m+1}^{d} \tilde{\lambda}_j (1 - \log \beta_j) + \tilde{\lambda}(1 - \prod_{j=m+1}^{d} \frac{\tilde{\lambda}_j}{\beta_j}). \tag{15}$$

Then there exist $\beta_j, j \neq j^*$ so that equation 15 matches $-\tilde{\lambda} \log n + \log C(d)$. $\qquad\square$

Theorem 5.1 critically assumes that $b(\xi) \propto 1$. We do not expect this to hold in reality. For example, even for the simple one-hidden layer $\tanh$ network considered in Section 7, it does not appear that $b(\xi) \propto 1$ (Watanabe, 2000). Currently, we are prevented from stating a more general version of Theorem 5.1 because there is no off-the-shelf derivation of the leading coefficient when the singular integral is of the general form $\int_{[0,b]^d} \exp(-n\xi^{2\tilde{\boldsymbol{k}}})\xi^{\tilde{\boldsymbol{h}}} b(\xi)\, d\xi$. We expect the generalization to be technically feasible but as its development requires advanced knowledge of algebraic geometry, it is beyond the scope of this paper and best left as separate investigation.

In the next section, we proceed to learn the resolution map rather than presume it is known. This is an improvement over Bhattacharya et al. (2020) which, due to the difficulty of deriving resolution maps, was limited in its single experiment to the toy neural network $f_w(x) = b\tanh(ax)$ with weight $w = (a, b) \in \mathbb{R}^2$. Again, though Theorem 5.1 is stated in terms of the truncated generalized gamma mean-field family, we do not believe the assumption $b(\xi) \propto 1$ is critical and **proceed henceforth to work with the *untruncated* generalized gamma mean-field family, which we denote $\mathcal{Q}$.**

## 6 LEARNING TO DESINGULARIZE

When the resolution map $g$ is known, the preceding results suggest to 1) apply the change-of-variables $g(\xi) = w$ and 2) maximize $\mathrm{ELBO}(q, g)$ over the untruncated generalized gamma mean-field family $\mathcal{Q}$ via e.g., stochastic gradient descent. However, theoretically, the resolution map $g$ is notoriously difficult to derive. Computationally, the recursive blow-up procedure in algebraic geometry is entirely not scalable to high dimensions.

We propose to learn the resolution map via a normalizing flow, which is commonly used to model complex distributions as the push-forward of a simple source distribution through an invertible neural network $G$. An interesting direction of future work would be to exploit properties of the resolution

map that may aid in the design of the normalizing flow architecture. For now, we simply make use of a common type of invertible architecture consisting of affine coupling layers. With $r$ denoting a binary mask, a so-called affine coupling layer acts as follow:

$$u, v \in \mathbb{R}^d, u \mapsto v = (1 - r) \odot u + r \odot (u \odot \exp(s(r \odot u)) + t(r \odot u)).$$

The binary mask $r$ must alternate from one affine coupling layer to the next for otherwise there would be little expressive power in the resulting network. Let $G_\theta$ be a network consisting of alternating affine coupling layers, where $\theta$ denotes the collective parameters. Since the resolution map is to be learned, there is no loss in generality to assuming $j^* = 1$. We shall need the gradient with respect to the variational parameters and the normalizing flow weights as part of employing stochastic gradient descent, i.e. $\nabla_{\boldsymbol{\lambda}, \boldsymbol{k}, \boldsymbol{\beta}, \theta} \mathrm{ELBO}(q_{\boldsymbol{\lambda}, \boldsymbol{k}, \boldsymbol{\beta}}, G_\theta)$. Note that here we abuse the notation slightly as we do not need to update $\beta_1$ which should be set to the sample size $n$ according to the proof of Theorem 5.1.

Although the source distribution in a normalizing flow can have its own trainable parameters, it is common practice to adopt a parameter-less source distribution. In particular if the source distribution is itself reparametrizable, then the learning of the associated parameters can be absorbed into the invertible transformation. The generalized gamma distribution is not easily reparametrizable for general values of $\boldsymbol{\lambda}, \boldsymbol{k}, \boldsymbol{\beta}$. However for certain settings of $\boldsymbol{\lambda}$, we can in fact avail ourselves to the reparametrization trick, at least approximately. Let $V_j$ be a gamma random variable with shape $\lambda_j$ and rate $\beta_j$, then $V_j^{1/(2k_j)} \sim q_j(\xi_j) := \xi_j^{2k_j \lambda_j - 1} \exp(-\beta_j \xi_j^{2k_j}) 1_{(0,\infty)}(\xi_j)$. We will mostly be interested in settings where $\lambda_j$ is large, in which case $V_j$ is approximately Gaussian with mean $\lambda_j / \beta_j$ and variance $\lambda_j / \beta_j^2$. Letting $T(\epsilon) := (F_1^{-1}(\epsilon_1)^{1/(2k_1)}, \dots, F_d^{-1}(\epsilon_d)^{1/(2k_d)})$, where $F_j^{-1}(\epsilon) \approx (\lambda_j + \sqrt{\lambda_j}\epsilon)/\beta_j$, the reparametrization trick then leads to the objective function

$$E_{\epsilon \sim N(0,I)} \left[ \sum_{i=1}^n \log p(y_i | x_i, G_\theta(T(\epsilon))) + \log \varphi(G_\theta(T(\epsilon))) + \log |G_\theta'(T(\epsilon))| \right] - E_q \log q. \quad (16)$$

Note that equation 16 is not the same as $\mathrm{ELBO}(q_{\boldsymbol{\lambda}, \boldsymbol{k}, \boldsymbol{\beta}}, G_\theta)$ because we have made use of the Gaussian approximation for $V_j$ in the case of large $\lambda_j$.

The entropy component of equation 16, $-E_{q_{\boldsymbol{\lambda}, \boldsymbol{k}, \boldsymbol{\beta}}} \log q_{\boldsymbol{\lambda}, \boldsymbol{k}, \boldsymbol{\beta}}$, can be derived analytically, see Appendix B.2. Next, the specific architecture of $G_\theta$ has rendered the log Jacobian term, $\log |G_\theta'(\cdot)|$, computationally tractable. The final piece is to replace $E_{\epsilon \sim N(0,I)}$ with an empirical average over $M$ samples. In the experiments that follow we will consider either learning the variational parameters $\boldsymbol{\lambda}, \boldsymbol{k}, \boldsymbol{\beta}$ or fixing them at some initial value since other values can be learned through a transformation that gets absorbed into $G$.

## 7 EXPERIMENTS

In this section, we compare the effect of two source distributions for a normalizing flow given by the affine coupling network $G_\theta$ consisting of 2 pairs of alternating couplings with scaling and translation networks each consisting of 2 hidden layers with 16 hidden units, see Appendix D for exact details of the architecture. We denote by `nf_gamma_`$\lambda_0$`_`$k_0$`_`$\beta_0$`_flag` the variational family that results from pushing forward the untruncated generalized gamma distribution with variational parameters initialized at $\boldsymbol{\lambda}_0 = (1, \lambda_0, \dots, \lambda_0)$, $\boldsymbol{k}_0 = (k_0, \dots, k_0)$, and $\boldsymbol{\beta}_0 = (n, \beta_0, \dots, \beta_0)$, and a boolean flag indicating whether the variational parameters are subject to updating. For the other approximation resulting from pushing forward a Gaussian source distribution $N(\mu_0, v_0)$, we write analogously `nf_gaussian_`$\mu_0$`_`$v_0$ where $\mu_0$ and $v_0$ are the fixed mean and variance. For each combination of $\lambda_0$ and $\beta_0$ considered, we set $\mu_0 = \lambda_0/\beta_0$ and $v_0 = \lambda_0/\beta_0^2$.

We employ the widely adopted variant of (minibatch) stochastic gradient descent known as Adam. The number of epochs and batch size were set to 2000 and $n/10$, respectively. Different constant learning rates were employed according to the parameter type: $1\mathrm{e}{-}3$ for affine coupling layer weights and $\boldsymbol{k}$, and $1\mathrm{e}{-}1$ for $\boldsymbol{\lambda}$ and $\boldsymbol{\beta}$. Expectations in the objective function that are not analytically tractable are replaced with an average over $M = 5$ samples. At the end of training, we evaluate $\Psi(q, g)$ in `nf_gamma` without recourse to the (approximate) reparametrization and by using the analytic expression for the entropy component of $\Psi$ and 100 samples from $q$ to approximate other components in $\Psi$ under expectation.

To date, there is a very small collection of model-truth-prior triplets where the RLCT $\tilde{\lambda}$ and multiplicity $m$ are known. We shall limit our experiments to two such triplets. This will allow us to compare the achieved $\Psi(\hat{q}, \hat{G})$ following training to $-\lambda \log n + (m-1) \log \log n$. Note that even in such triplets where the RLCT and multiplicity are known, the exact value of the leading coefficient $C$ is still usually unknown. Now, to compare `nf_gamma` and `nf_gaussian` to each other, we can simply see which achieves higher $\Psi(\hat{q}, \hat{G})$ after training.

Looking ahead to downstream tasks, it is natural to ask whether the variational posterior predictive distribution, $p_{vb}(y|x, \mathcal{D}_n) = \langle p(y|x, G_{\hat{\theta}}(\xi)) \rangle_{\hat{q}(\xi)}$, inherits the desirable properties of the Bayes posterior predictive distribution, $p(y|x, \mathcal{D}_n) = \langle p(y|x, w) \rangle_{p(w|\mathcal{D}_n)}$. The answer turns out to depend on the relationship between the **variational real log canonical threshold** $\lambda_{vb}$ and the RLCT of the model-truth-prior triplet. It may very well be that a variational family which is closer to the true posterior (in the KL sense) than another variational family may induce a *worse* approximation of the true Bayes posterior predictive distribution. We provide an in-depth account of this phenomenon through the lens of singular learning theory in Appendix E.

Our first example concerns the tanh model-truth-prior triplet. Consider input $x \in \mathbb{R}$ following the uniform distribution on $[-1, 1]$, and response variable $y \in \mathbb{R}$ modeled as $p(y|x, w) = \frac{1}{\sqrt{2\pi}} \exp(-\frac{1}{2}(y - f(x, w))^2)$, where $f_w(x) = \sum_{h=1}^{H} b_h \tanh(a_h x)$ is a tanh network with $H$ hidden units and $w$ is the collection of neural network weights $\{(a_h, b_h)\}_{h=1}^{H}$. If the true distribution is given by $p_0(y|x) = p(y|x, 0) = \frac{1}{\sqrt{2\pi}} \exp(-\frac{1}{2}y^2)$ and the prior $\varphi$ is a $C^\infty$ function of $w$ with compact support, satisfying $\varphi(0) > 0$, Aoyagi & Watanabe (2006) showed that $\lambda = \frac{H + i^2 + i}{4i + 2}$ and $m = 2$ if $i^2 = H$, and $m = 1$ if $i^2 < H$ where $i$ is the maximum integer satisfying $i^2 \leq H$. In contrast, were this a regular statistical model, we would have $\lambda = H$. We consider two settings for $p_0(y|x) = p(y|x, w_0)$ in the experiment: 1) $w_0 = 0$ and 2) $w_0 = 5$. Note for the latter, the corresponding RLCT and multiplicity are unknown.

Our next example is the reduced rank model-truth-prior triplet. Consider input $x \in \mathbb{R}^M$ and response variable $y \in \mathbb{R}^N$ modeled as $p(y|x, w) = (2\pi)^{-N/2} \exp\{-\frac{1}{2}||y - BAx||^2\}$ where $\{w = (A, B) | A \in \mathbb{R}^{H \times M}, B \in \mathbb{R}^{N \times H}\}$. This model is readily seen to be a special case of a neural network with hidden units $H$ and identity activation function. We shall set $M = H + 3$ and $N = H$. In the realizable case, i.e., $p_0(y|x) = p(y|x, A_0, B_0)$, if the prior $\varphi$ is a $C^\infty$ function with compact support satisfying $\varphi(A_0, B_0) > 0$, the RLCT was derived in Aoyagi & Watanabe (2005) for various values of $N, M, H$ and $r = \text{rank}(B_0 A_0)$. Below we set $B_0 = I_{N \times N}$ and $A_0 = [I_{H \times H}; J_{H \times 3}]$. The rank $r$ for $B_0 A_0$ equals $H$. This then falls under Case (3) in Aoyagi & Watanabe (2005) since $N + H < M + r$, leading to $\lambda = (NH - Hr + Mr)/2, m = 1$. Note that were this a regular model, we would instead have $\lambda = (MH + NH)/2$.

Table 1 provides a summary of the values of $H$ considered in each of the triplets and the corresponding RLCT and dimension. In the experiments, we tested various combinations of values of $H$, settings for $w_0$ and priors. Throughout, the priors were designed to be mis-specified. The full results of these experiments can be found in Tables 2–4. Note that the comparison between different variational approximations have been made subject to the value of $H$, the prior, and the relationship $\lambda_0/\beta_0 = \mu_0$. We should also point out that not all runs reached convergence as can be seen from the count column in Tables 2–4. This seems largely to be a result of the constant learning rate we used for all training.

The **first setting** (Table 2) is the realizable tanh network with $w_0 = 0$. In this case, the true RLCT is known as discussed above. As Table 2 demonstrates, there is no clearly discernible difference between `nf_gamma` and `nf_gaussian` where the source distribution for both starts near zero. However when the source distribution has $\lambda_0/\beta_0 = \mu_0 = 5$, i.e. very far from $w_0$, `nf_gamma` is more robust than `nf_gaussian`. The **second setting** (Table 3) is analogous to the previous except the true generating mechanism is given by $w_0 = 5$. In this setting, the untruncated generalized gamma source distribution is seen to provide a better variational approximation than the Gaussian source distribution. This appears to hold quite independent of the initial values of $\boldsymbol{\lambda}, \boldsymbol{k}, \boldsymbol{\beta}$ as well as the gradient flag. Interestingly, in contrast to the previous setting, there is a clear benefit to learning $\boldsymbol{\lambda}, \boldsymbol{k}, \boldsymbol{\beta}$. The **third setting** (Table 4) is the realizable reduced rank regression. In this experiment, we do not see a discernible difference between the source distributions until we look at $\lambda_0/\beta_0 = \mu_0 = 5$, which is far from $w_0 = (A_0, B_0)$. As in the first setting, we see `nf_gamma` is more robust than `nf_gaussian`.

Finally, though we have cautioned against judging the quality of a variational approximation according to the approximate posterior predictive distribution induced, it is nonetheless informative to visualize downstream uncertainty quantification. In Figure 1, we choose one line from the tanh $w_0 = 0$ experiment (Table 2) to further examine. For the two methods illustrated in in Figure 1, their performance in terms of $\Psi$ is quite close as indicated by Table 2. It is then comforting that the respective confidence bands are quite similar. Next, in Figure 2, we choose one line from the tanh $w_0 = 5$ experiment (Table 3) to further examine. This time, `nf_gamma` achieves significantly better $\Psi$ than `nf_gaussian`. We see a corresponding relationship between their respective posterior predictive distributions. Namely, the confidence band resulting from `nf_gaussian` are far too conservative while that from `nf_gamma` is less so.

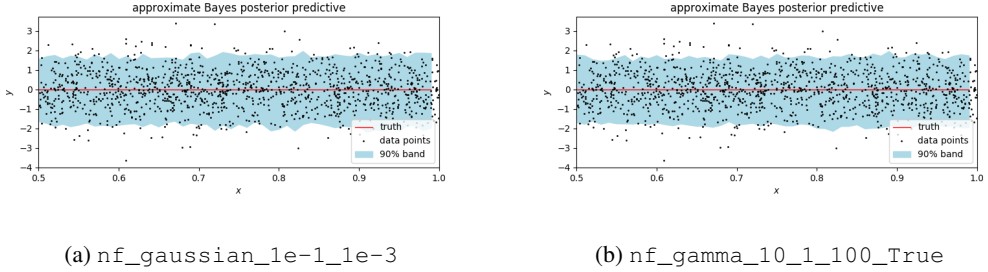

(a) `nf_gaussian_1e-1_1e-3`  (b) `nf_gamma_10_1_100_True`

Figure 1: The model of interest is the tanh network with hidden units $H = 576$. The data is generated according to $p_0(y|x, w) = p(y|x, w_0)$ where $w_0 = 0$. The prior is taken to be $N(5.0, 100.0 I_d)$. The predictive distributions resulting from different variational approximations trained on a dataset of size $n = 5000$ are displayed.

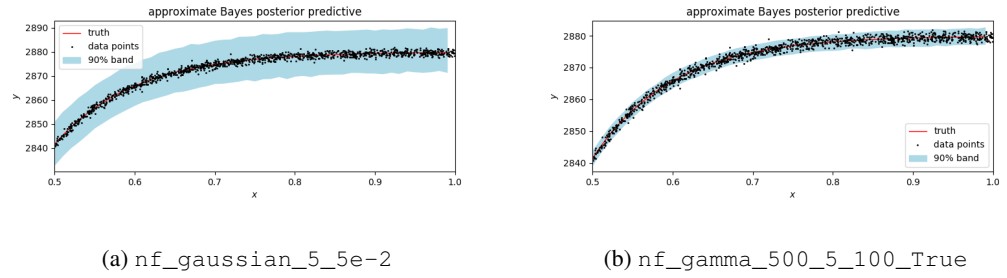

(a) `nf_gaussian_5_5e-2`  (b) `nf_gamma_500_5_100_True`

Figure 2: The model of interest is the tanh network with hidden units $H = 576$. The data is generated according to $p_0(y|x, w) = p(y|x, w_0)$ where $w_0 = 5$. The prior is taken to be $N(0.0, 100.0 I_d)$. The predictive distributions resulting from different variational approximations trained on a dataset of size $n = 5000$ are displayed.

## 8 CONCLUSION

In this work we propose a variational approximation for Bayesian neural networks by leveraging insights from singular learning theory. Namely, for large $n$, the posterior distribution over neural network weights is not Gaussian but rather can be put into a mixture of standard forms. From this, we demonstrate that the generalized gamma mean-field family, following desingularization, can in theory achieve the leading order term of the log normalized evidence. Because we choose to learn the desingularization map using affine coupling layers, the proposed work can be cast as a normalizing flow with an unconventional source distribution. Interestingly, for large values of the variational parameter $\boldsymbol{\lambda}$, the source distribution in each dimension is approximately $N(\lambda, \lambda/\beta)^{1/(2k)}$, which is reparametrizable in terms of the conventional source distribution $N(0, 1)$. Though learning the variational parameters $\boldsymbol{\lambda}, \boldsymbol{k}, \boldsymbol{\beta}$ in the source distribution goes against conventional wisdom in normalizing flows, our experiments suggest some performance may be gained by learning the optimal untruncated generalized gamma source distribution at the same time as learning the resolution map.

ETHICS STATEMENT

Bayesian learning for neural networks is often touted as a panacea to the challenges of uncertainty quantification in deep learning. In this work, we propose a variational approximation that performs well *in terms of the ELBO achieved*. However, we make no claim that our variational approximation is superior in downstream uncertainty quantification. We explain extensively in the appendix that a variational approximation which achieves *higher* ELBO may induce a *worse* approximate posterior predictive distribution than a variational approximation which achieves lower ELBO. According to singular learning theory, the performance of a variational approximation in terms of the posterior predictive distribution it induces critically depends on the relationship between the variational RLCT and the underlying RLCT of the model-truth-prior triplet.

REPRODUCIBILITY STATEMENT

A zip file of the source code has been submitted as supplementary materials. For theoretical results, explanations of assumptions and proof of the claims are included in the main text. No datasets are used; the experiments are solely based on simulated data.

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

## A  TOY EXAMPLE OF RLCT CALCULATION

We recall Example 27 from Watanabe (2018) to illustrate the concepts of resolution map, RLCT and multiplicity for a simple model-truth-prior triplet. For univariate input $x \in [0, 1]$ and univariate output $y \in \mathbb{R}$, consider the model with parameter $w = (a, b) \in [0, 1]^2$ given by

$$p(x, y|w) = \frac{1}{\sqrt{2\pi}} \exp(-\frac{1}{2}(y - a\tanh(bx))^2) \tag{17}$$

Suppose the prior is uniform, i.e., $\varphi(w) = 1$ and the truth is given by $p_0(x, y) = p(x, y|0, 0)$. Then we can easily see that

$$K(w) = b^2 a^2 \frac{1}{2} K_0(w),$$

where

$$K_0(w) = \int_0^1 \left(\frac{\tanh(bx)}{b}\right)^2 dx$$

The following desingularization map puts the triplet in standard form:

$$\xi_1 = \sqrt{\frac{K_0(w)}{2}} a$$
$$\xi_2 = b$$

Furthermore we have $\varphi(g(\xi)) = \xi^h$ where $h = (0, 0)$ and $b(\xi) = |g'(\xi)|$. Since $(k_1, k_2) = (1, 1)$ and $(h_1, h_2) = (0, 0)$ we have $(\lambda_1, \lambda_2) = (1/2, 1/2)$. Therefore for this particular model-truth-prior triplet, the RLCT is $1/2$ with multiplicity 2.

## B  BASIC CALCULATIONS FOR THE GENERALIZED GAMMA DISTRIBUTION

### B.1  TRUNCATED

We summarize some basic calculations for the $[0, 1]$ truncated generalized gamma distribution from Bhattacharya et al. (2020). Let $q_j$ be the univariate truncated generalized gamma density given in equation 12.

- The normalizing constant of $q_j$ is given by $B(\lambda_j, k_j, \beta_j)$ where $B(\lambda, k, \beta) = \frac{\beta^{-\lambda}\Gamma(\lambda)\gamma(\lambda,\beta)}{2k}$ and $\gamma(a, x) = \frac{1}{\Gamma(a)} \int_0^x t^{a-1}e^{-t}\, dt$ is the (regularized) lower incomplete gamma function.

- The quantity $E_{q_j}\xi^{2k_j} = G(\lambda_j, \beta_j)$ where $G(\lambda, \beta) = \frac{\lambda}{\beta}\frac{\gamma(\lambda+1,\beta)}{\gamma(\lambda,\beta)}$.

### B.2  UNTRUNCATED

Consider the univariate density

$$q(\xi) \propto \xi^h \exp(-\beta\xi^{2k})$$

supported on $(0, \infty)$ where $h, k, \beta > 0$. Let $\lambda = (h + 1)/(2k)$. Elementary calculations give

- $E_q\xi^{2\tilde{k}} = \beta^{-\frac{\tilde{k}}{k}}\frac{\Gamma(\frac{\tilde{k}}{k}+\lambda)}{\Gamma(\lambda)}$

- $E_q \log \xi = \frac{1}{2k}(\psi(\lambda) - \log\beta)$ where $\psi$ is the digamma function.

- $E_q \log q = \frac{h}{2k}(\psi(\lambda) - \log\beta) - \lambda - \log Z$ where, $Z = \frac{\beta^{-\lambda}\Gamma(\lambda)}{2k}$ is the normalizing constant of $q$.

## C  GENERALIZATION OF (BHATTACHARYA ET AL., 2020, THEOREM 3.1)

Lemma C.1 below states that any variational distribution $q_{\boldsymbol{\lambda},\boldsymbol{k},\boldsymbol{\beta}}$ in the untruncated generalized gamma mean field family $\mathcal{Q}$ with well-specified $\boldsymbol{k}$ where we additionally set $\beta_{j^*} = n$ is capable of achieving $\Psi_K(q, g) \geq -\tilde{\lambda}\log n + A$ where $A$ is some constant that does not depend on $n$. Lemma C.1 is a straightforward extension of Theorem 3.1 of Bhattacharya et al. (2020)from the $[0, 1]^d$ truncated generalized gamma mean field variational family to the untruncated case. We also relax the condition in Theorem 3.1 of Bhattacharya et al. (2020) that the variational parameters $\boldsymbol{\lambda}$ and $\boldsymbol{k}$ (and hence $\boldsymbol{h}$) are well-specified; here we only require that $\boldsymbol{k}$ be well-specified.

**Lemma C.1.** *Suppose the model-truth-prior triplet is such that Theorem 3.1 holds with $K(g(\xi)) = \xi^{2\tilde{\boldsymbol{k}}}$ and $\varphi(g(\xi))|g'(\xi)| = \xi^{\tilde{\boldsymbol{h}}}$. Consider the untruncated generalized gamma mean-field family $\mathcal{Q}$, we have $\sup_{q \in \mathcal{Q}} \Psi_K(q, g) \geq -\tilde{\lambda}\log n + A$, where $A$ is a constant that does not depend on $n$.*

*Proof.* If $k_j = \tilde{k}_j$ for all $j = 1, \ldots, d$, we have

$$E_q nK(g(\xi)) = n\prod_{j=1}^d \beta_j^{-\frac{\tilde{k}_j}{k_j}}\frac{\Gamma(\frac{\tilde{k}_j}{k_j}+\lambda_j)}{\Gamma(\lambda_j)} = n^{1-\frac{\tilde{k}_{j^*}}{k_{j^*}}}\frac{\Gamma(\frac{\tilde{k}_{j^*}}{k_{j^*}}+\lambda_{j^*})}{\Gamma(\lambda_{j^*})}\prod_{j\neq j^*}\beta_j^{-\frac{\tilde{k}_j}{k_j}}\frac{\Gamma(\frac{\tilde{k}_j}{k_j}+\lambda_j)}{\Gamma(\lambda_j)} = \lambda_{j^*}\prod_{j\neq j^*}\frac{\lambda_j}{\beta_j}.$$

Next, we have

$$\mathrm{KL}(q(\xi)||\varphi(g(\xi))|g'(\xi)|)$$

$$= \sum_{j=1}^{d} E_{q_j}[\log q_j - \tilde{h}_j \log \xi_j] - E_q \log b(\xi)$$

$$= \sum_{j=1}^{d} \left[ \frac{h_j - \tilde{h}_j}{2k_j}(\psi(\lambda_j) - \log \beta_j) - \lambda_j + \lambda_j \log \beta_j + \log(2k_j) - \log \Gamma(\lambda_j) \right] - E_q \log b(\xi)$$

$$= \sum_{j=1}^{d} \left[ \frac{h_j - \tilde{h}_j}{2k_j}\psi(\lambda_j) - \lambda_j + \frac{\tilde{h}_j + 1}{2k_j} \log \beta_j + \log(2k_j) - \log \Gamma(\lambda_j) \right] - E_q \log b(\xi)$$

$$= \sum_{j=1}^{d} \left[ \frac{h_j - \tilde{h}_j}{2\tilde{k}_j}\psi(\lambda_j) - \lambda_j + \tilde{\lambda}_j \log \beta_j + \log(2\tilde{k}_j) - \log \Gamma(\lambda_j) \right] - E_q \log b(\xi)$$

$$= \tilde{\lambda} \log n + \sum_{j \neq j^*} \tilde{\lambda}_j \log \beta_j + \sum_{j=1}^{d} \left[ \frac{h_j - \tilde{h}_j}{2\tilde{k}_j}\psi(\lambda_j) - \lambda_j + \log(2\tilde{k}_j) - \log \Gamma(\lambda_j) \right] - E_q \log b(\xi).$$

The last line follows from setting $\beta_{j^*} = n$ where $j^* \in \{1, \ldots, d\}$ is such that $\tilde{\lambda}_{j^*} = \tilde{\lambda}$, breaking ties arbitrarily. Finally, the desired inequality follows from the fact that $E_q \log b(\xi)$ is bounded below by some constant. $\qquad\square$

## D EXPERIMENTS

Table 1: Summary of models considered.

| $H$ | RLCT | $d$ |
|-----|------|-----|
| 576 | 12.0 | 1152 |
| 1024 | 16.0 | 2048 |

(a) $\tanh$ network

| $H$ | RLCT | $d$ |
|-----|------|-----|
| 24 | 324.0 | 1224 |
| 32 | 560.0 | 2144 |

(b) Reduced rank

In the experiments, we train an affine coupling network with two pairs of alternating couplings. The translation $t$ is a feedforward (leaky) ReLU neural network with $\tanh$ output activation function. The scaling $t$ is another feedforward (leaky) ReLU neural network with identity output activation function. The models implemented are given below.

Translation network

```
Sequential(
(0): Linear(in_features=dim, out_features=16, bias=True)
(1): LeakyReLU(negative_slope=0.01)
(2): Linear(in_features=16, out_features=16, bias=True)
(3): LeakyReLU(negative_slope=0.01)
(4): Linear(in_features=16, out_features=16, bias=True)
(5): LeakyReLU(negative_slope=0.01)
(6): Linear(in_features=16, out_features=dim, bias=True)
)
```

Scaling network

```
Sequential(
(0): Linear(in_features=dim, out_features=16, bias=True)
(1): LeakyReLU(negative_slope=0.01)
(2): Linear(in_features=16, out_features=16, bias=True)
(3): LeakyReLU(negative_slope=0.01)
(4): Linear(in_features=16, out_features=16, bias=True)
(5): LeakyReLU(negative_slope=0.01)
(6): Linear(in_features=16, out_features=dim, bias=True)
(7): Tanh()
)
```

# E  APPROXIMATE POSTERIOR PREDICTIVE DISTRIBUTION

Bayesian prediction proceeds by marginalization, i.e., averaging over all possible values of the model parameter. This results in the posterior predictive distribution[1],

$$p(y|x, \mathcal{D}_n) := \int_w p(y|x, w)p(w|\mathcal{D}_n)\, dw = \langle p(y|x, w) \rangle_{p(w|\mathcal{D}_n)}. \tag{18}$$

According to singular learning theory, Bayesian prediction via equation 18 is superior to MAP and MLE, in the sense that the expected generalization error of the posterior predictive distribution is smaller than that of the MAP or MLE. Specifically, let

$$G_n(\hat{p}_n(y|x)) := \mathrm{KL}(p_0(y|x)p(x)||\hat{p}_n(y|x)p(x))$$

be the generalization error of a predictive distribution $\hat{p}_n(y|x)$. According to Theorems 1.2 and 7.2 in Watanabe (2009), we have

$$EG_n(p(y|x, \mathcal{D}_n)) = \lambda/n + o(1/n) \tag{19}$$

where the expectation is taken with respect to $\mathcal{D}_n$ and $\lambda$ is the RLCT of the model-truth-prior triplet. On the other hand, Theorem 6.4 of Watanabe (2009) shows that the expected generalization error of MLE (and similarly of MAP) is

$$EG_n(p(y|x, \hat{w}_{mle})) = C/n + o(1/n) \tag{20}$$

where $C$, the maximum of a Gaussian process, can be much larger than $\lambda$. Such a distinction cannot be made in regular models in which the difference between the three estimators becomes negligible in the large $n$ regime.

Now, consider the variational approximation to the posterior predictive distribution given by

$$p_{vb}(y|x, \mathcal{D}_n) = \langle p(y|x, w) \rangle_{q^*(w)},$$

where $q^*$ is the optimal variational distribution in some variational family $\mathcal{Q}$. It may be tempting to compare different variational approximations according to how well their respective variational posterior predictive distributions approximate the true Bayes posterior predictive distribution, e.g., Blundell et al. (2015); Louizos & Welling (2017). This turns out to be a thorny issue, as documented in various works on Bayesian neural networks Heek (2018); Krishnan & Tickoo (2020); Foong et al. (2020). In particular, $p_{vb}(y|x, \mathcal{D}_n)$ does not necessarily inherit the desirable properties of the Bayes posterior predictive distribution $p(y|x, \mathcal{D}_n)$. Here, we offer insights from singular learning theory to account for this.

Consider the (normalized) variational free energy

$$\bar{F}_{vb}(n) = E_q n K_n(w) + \mathrm{KL}(q(w)||\varphi(w))$$

where $q$ is some variational distribution. If the minimum (normalized) variational free energy $\bar{F}_{vb}^*(n) = \min_q \bar{F}_{vb}(n)$ admits an asymptotic expansion[2], then it would be of the form

$$\bar{F}_{vb}^*(n) = \lambda_{vb} \log n + (m_{vb} - 1) \log \log n + R_n.$$

Note that $\lambda_{vb} \geq \lambda$ necessarily. Now, if the generalization error of $p_{vb}(y|x, \mathcal{D}_n)$ would admit an asymptotic expansion[3], it would be of the form

$$EG_n(p_{vb}(y|x, \mathcal{D}_n)) = \tilde{\lambda}_{vb}/n + o(1/n).$$

---

[1]Though this may seem fundamentally distinct from the maximum likelihood estimator (MLE) or maximum a posterior (MAP) solution commonly employed in training deep networks, both MLE and MAP may in fact be regarded as impoverished estimates of equation 18 whereby $p(w|\mathcal{D}_n)$ is approximated with $\delta(w = \hat{w})$, a point mass at $\hat{w}$.

[2]We should disclose that general conditions for such an asymptotic expansion of the minimum variational free energy is still an open problem. The issue has so far been addressed on a case-by-case basis, e.g., reduced rank regression Nakajima & Watanabe (2007), nonnegative matrix factorization Kohjima & Watanabe (2017); Hayashi (2020), normal mixture model Watanabe & Watanabe (2006), hidden Markov model Hosino et al. (2005).

[3]Similar to the minimum variational free energy, an expansion of $EG_n(p_{vb}(y|x, \mathcal{D}_n))$ has not been established in full generality at this point.

Importantly, $\tilde{\lambda}_{vb} \neq \lambda_{vb}$ in general. Examples can be found where sometimes one is bigger, sometimes the other Nakajima & Watanabe (2007).

This development is to be contrasted with the Bayes free energy and the Bayes generalization error. Recall the normalized Bayes free energy, $\bar{F}_n := -\log \bar{Z}(n)$, admits the following asymptotic expansion

$$\bar{F}_n = \lambda \log n + (m-1) \log \log n + R_n. \tag{21}$$

The very same $\lambda$ coefficient appears in the asymptotic expansion of the Bayes generalization error in equation 19. This means that minimizing the Bayes free energy is equivalent to minimizing the Bayes generalization error. In contrast, minimizing the variational free energy (equivalent to maximizing the ELBO) does not necessarily lead to a lower generalization error since a variational family with higher variational free energy (higher $\lambda_{vb}$) may have lower variational generalization error (lower $\tilde{\lambda}_{vb}$).

Table 2: The model of interest is the realizable $\mathtt{tanh}$-network with $H$ hidden units and true weight $w_0 = 0$. The prior on the model parameters is taken to be $\varphi(w) = N(\mu(\varphi), \sigma^2(\varphi)I_d)$. For various combinations of values of $H$ and values for $(\mu(\varphi), \sigma^2(\varphi))$, we display the average and standard deviation of $\Psi(q, g)$ achieved after training on a dataset of size $n = 5000$ over 10 Monte Carlo iterations.

| $H$ | $(\mu(\varphi), \sigma^2(\varphi))$ | $\lambda_0/\beta_0 = \mu_0$ | method | count | mean | std |
|---|---|---|---|---|---|---|
| | | 0.00 | nf_gaussian_0_1 | 10.0 | -12307.14 | 1153.17 |
| | | | nf_gamma_10_1_100_False | 10.0 | -10894.83 | 836.64 |
| | | | nf_gamma_10_1_100_True | 10.0 | -10768.14 | 894.35 |
| | | 0.10 | nf_gamma_10_5_100_False | 10.0 | -12476.67 | 877.12 |
| | | | nf_gamma_10_5_100_True | 10.0 | -11849.96 | 415.19 |
| | | | nf_gaussian_0.1_0.001 | 10.0 | -10604.66 | 528.57 |
| | | | nf_gamma_100_1_100_False | 10.0 | -12804.39 | 1303.57 |
| | (5.0, 1.0) | | nf_gamma_100_1_100_True | 10.0 | -11759.06 | 610.11 |
| | | 1.00 | nf_gamma_100_5_100_False | 10.0 | -14328.32 | 1215.64 |
| | | | nf_gamma_100_5_100_True | 10.0 | -13079.31 | 1022.25 |
| | | | nf_gaussian_1_1e-2 | 10.0 | -11658.58 | 713.89 |
| | | | nf_gamma_500_1_100_False | 10.0 | -16915.87 | 2048.86 |
| | | | nf_gamma_500_1_100_True | 10.0 | -17939.35 | 6950.49 |
| | | 5.00 | nf_gamma_500_5_100_False | 10.0 | -15146.53 | 639.40 |
| | | | nf_gamma_500_5_100_True | 10.0 | -15079.50 | 974.71 |
| 576 | | | nf_gaussian_5_5e-2 | 10.0 | -20300.63 | 2274.93 |
| | | 0.00 | nf_gaussian_0_1 | 10.0 | -4694.71 | 42.93 |
| | | | nf_gamma_10_1_100_False | 10.0 | -4919.57 | 85.38 |
| | | | nf_gamma_10_1_100_True | 10.0 | -4928.69 | 75.24 |
| | | 0.10 | nf_gamma_10_5_100_False | 10.0 | -6036.98 | 511.80 |
| | | | nf_gamma_10_5_100_True | 9.0 | -5909.62 | 116.19 |
| | | | nf_gaussian_0.1_0.001 | 10.0 | -4722.43 | 53.72 |
| | | | nf_gamma_100_1_100_False | 10.0 | -5460.35 | 236.25 |
| | (5.0, 100.0) | | nf_gamma_100_1_100_True | 10.0 | -5060.05 | 186.63 |
| | | 1.00 | nf_gamma_100_5_100_False | 10.0 | -7590.89 | 254.81 |
| | | | nf_gamma_100_5_100_True | 10.0 | -6098.62 | 307.36 |
| | | | nf_gaussian_1_1e-2 | 10.0 | -8010.75 | 9394.81 |
| | | | nf_gamma_500_1_100_False | 10.0 | -7144.50 | 452.06 |
| | | | nf_gamma_500_1_100_True | 10.0 | -6842.81 | 199.32 |
| | | 5.00 | nf_gamma_500_5_100_False | 10.0 | -8392.05 | 207.44 |
| | | | nf_gamma_500_5_100_True | 9.0 | -8475.71 | 412.61 |
| | | | nf_gaussian_5_5e-2 | 10.0 | -10974.67 | 4667.23 |
| | | 0.00 | nf_gaussian_0_1 | 10.0 | -24543.92 | 695.44 |
| | | | nf_gamma_10_1_100_False | 10.0 | -20607.23 | 1776.76 |
| | | | nf_gamma_10_1_100_True | 5.0 | -19642.01 | 449.75 |
| | | 0.10 | nf_gamma_10_5_100_False | 9.0 | -25668.64 | 1783.73 |
| | | | nf_gamma_10_5_100_True | 9.0 | -25693.87 | 2344.70 |
| | | | nf_gaussian_0.1_0.001 | 10.0 | -20014.07 | 505.24 |
| | | | nf_gamma_100_1_100_False | 10.0 | -26573.95 | 1378.88 |
| | (5.0, 1.0) | | nf_gamma_100_1_100_True | 10.0 | -26259.66 | 2921.10 |
| | | 1.00 | nf_gamma_100_5_100_False | 9.0 | -28239.06 | 1348.93 |
| | | | nf_gamma_100_5_100_True | 9.0 | -29061.34 | 2843.18 |
| | | | nf_gaussian_1_1e-2 | 10.0 | -24311.66 | 2924.31 |
| | | | nf_gamma_500_1_100_False | 10.0 | -35135.85 | 10110.49 |
| | | | nf_gamma_500_1_100_True | 10.0 | -34346.62 | 7563.82 |
| | | 5.00 | nf_gamma_500_5_100_False | 10.0 | -31770.45 | 1765.41 |
| | | | nf_gamma_500_5_100_True | 8.0 | -31673.14 | 2167.54 |
| 1024 | | | nf_gaussian_5_5e-2 | 10.0 | -243192.15 | 271227.44 |
| | | 0.00 | nf_gaussian_0_1 | 10.0 | -8396.27 | 62.13 |
| | | | nf_gamma_10_1_100_False | 10.0 | -8705.24 | 77.06 |
| | | | nf_gamma_10_1_100_True | 10.0 | -8847.36 | 127.36 |
| | | 0.10 | nf_gamma_10_5_100_False | 9.0 | -11616.24 | 1014.68 |
| | | | nf_gamma_10_5_100_True | 8.0 | -10916.36 | 377.03 |
| | | | nf_gaussian_0.1_0.001 | 10.0 | -8425.56 | 71.42 |
| | | | nf_gamma_100_1_100_False | 10.0 | -10916.45 | 479.14 |
| | (5.0, 100.0) | | nf_gamma_100_1_100_True | 10.0 | -9514.43 | 397.57 |
| | | 1.00 | nf_gamma_100_5_100_False | 9.0 | -14374.43 | 1187.21 |
| | | | nf_gamma_100_5_100_True | 8.0 | -12994.13 | 765.11 |
| | | | nf_gaussian_1_1e-2 | 10.0 | -9408.79 | 480.26 |
| | | | nf_gamma_500_1_100_False | 10.0 | -21823.85 | 9739.67 |
| | | | nf_gamma_500_1_100_True | 10.0 | -44768.74 | 94773.50 |
| | | 5.00 | nf_gamma_500_5_100_False | 9.0 | -15257.90 | 282.52 |
| | | | nf_gamma_500_5_100_True | 8.0 | -16369.58 | 1287.13 |
| | | | nf_gaussian_5_5e-2 | 10.0 | -182386.08 | 155290.52 |

Table 3: The model of interest is the realizable $\mathtt{tanh}$-network with $H$ hidden units and true weight $w_0 = 5$. The prior on the model parameters is taken to be $\varphi(w) = N(\mu(\varphi), \sigma^2(\varphi)I_d)$. For various combinations of values of $H$ and values for $(\mu(\varphi), \sigma^2(\varphi))$, we display the average and standard deviation of $\Psi(q, g)$ achieved after training on a dataset of size $n = 5000$ over 10 Monte Carlo iterations.

| $H$ | $(\mu(\varphi), \sigma^2(\varphi))$ | $\lambda_0/\beta_0 = \mu_0$ | method | count | mean | std |
|---|---|---|---|---|---|---|
| 576 | (0.0, 1.0) | 0.00 | nf_gaussian_0_1 | 7.0 | -481210.04 | 290152.33 |
| | | | nf_gamma_10_1_100_False | 6.0 | -564637.97 | 185632.52 |
| | | | nf_gamma_10_1_100_True | 9.0 | -466874.65 | 229687.03 |
| | | 0.10 | nf_gamma_10_5_100_False | 10.0 | -189065.73 | 110357.18 |
| | | | nf_gamma_10_5_100_True | 10.0 | -66869.47 | 18069.35 |
| | | | nf_gaussian_0.1_0.001 | 9.0 | -308620.24 | 270119.27 |
| | | | nf_gamma_100_1_100_False | 10.0 | -190317.07 | 74299.40 |
| | | | nf_gamma_100_1_100_True | 10.0 | -130158.69 | 76731.88 |
| | | 1.00 | nf_gamma_100_5_100_False | 10.0 | -48865.52 | 14299.86 |
| | | | nf_gamma_100_5_100_True | 10.0 | -53566.72 | 18364.18 |
| | | | nf_gaussian_1_1e-2 | 10.0 | -544930.96 | 194349.17 |
| | | | nf_gamma_500_1_100_False | 10.0 | -89427.42 | 17721.97 |
| | | | nf_gamma_500_1_100_True | 10.0 | -81720.39 | 13133.21 |
| | | 5.00 | nf_gamma_500_5_100_False | 10.0 | -33854.50 | 8232.85 |
| | | | nf_gamma_500_5_100_True | 10.0 | -32275.02 | 7026.42 |
| | | | nf_gaussian_5_5e-2 | 10.0 | -100876.71 | 26177.59 |
| | (0.0, 100.0) | 0.00 | nf_gaussian_0_1 | 8.0 | -407843.61 | 96924.16 |
| | | | nf_gamma_10_1_100_False | 5.0 | -503729.89 | 39713.42 |
| | | | nf_gamma_10_1_100_True | 9.0 | -407507.24 | 172978.81 |
| | | 0.10 | nf_gamma_10_5_100_False | 10.0 | -154222.94 | 81217.70 |
| | | | nf_gamma_10_5_100_True | 10.0 | -64598.97 | 46518.60 |
| | | | nf_gaussian_0.1_0.001 | 9.0 | -313547.55 | 337434.22 |
| | | | nf_gamma_100_1_100_False | 10.0 | -170076.39 | 81803.27 |
| | | | nf_gamma_100_1_100_True | 10.0 | -112674.00 | 72282.16 |
| | | 1.00 | nf_gamma_100_5_100_False | 10.0 | -37469.45 | 24653.37 |
| | | | nf_gamma_100_5_100_True | 10.0 | -47135.05 | 34424.37 |
| | | | nf_gaussian_1_1e-2 | 10.0 | -523308.71 | 212480.52 |
| | | | nf_gamma_500_1_100_False | 10.0 | -69257.83 | 21314.18 |
| | | | nf_gamma_500_1_100_True | 10.0 | -58690.81 | 12706.57 |
| | | 5.00 | nf_gamma_500_5_100_False | 10.0 | -17097.24 | 6511.72 |
| | | | nf_gamma_500_5_100_True | 10.0 | -14548.94 | 4430.04 |
| | | | nf_gaussian_5_5e-2 | 10.0 | -87386.90 | 26349.19 |
| 1024 | (0.0, 1.0) | 0.00 | nf_gaussian_0_1 | 3.0 | -895765.67 | 77889.00 |
| | | | nf_gamma_10_1_100_False | 2.0 | -924993.41 | 100733.15 |
| | | | nf_gamma_10_1_100_True | 7.0 | -671229.30 | 160802.33 |
| | | 0.10 | nf_gamma_10_5_100_False | 8.0 | -558522.30 | 252813.58 |
| | | | nf_gamma_10_5_100_True | 10.0 | -279065.90 | 225130.01 |
| | | | nf_gaussian_0.1_0.001 | 6.0 | -452279.64 | 129732.34 |
| | | | nf_gamma_100_1_100_False | 10.0 | -345768.88 | 61677.23 |
| | | | nf_gamma_100_1_100_True | 10.0 | -213500.49 | 64295.75 |
| | | 1.00 | nf_gamma_100_5_100_False | 10.0 | -183015.46 | 128751.21 |
| | | | nf_gamma_100_5_100_True | 10.0 | -117581.92 | 42164.41 |
| | | | nf_gaussian_1_1e-2 | 6.0 | -836952.29 | 71929.66 |
| | | | nf_gamma_500_1_100_False | 10.0 | -196193.05 | 44549.27 |
| | | | nf_gamma_500_1_100_True | 10.0 | -179038.64 | 52969.12 |
| | | 5.00 | nf_gamma_500_5_100_False | 10.0 | -66127.35 | 6789.25 |
| | | | nf_gamma_500_5_100_True | 10.0 | -67303.85 | 17901.20 |
| | | | nf_gaussian_5_5e-2 | 10.0 | -216075.28 | 71401.02 |
| | (0.0, 100.0) | 0.00 | nf_gaussian_0_1 | 3.0 | -750263.23 | 140506.97 |
| | | | nf_gamma_10_1_100_False | 3.0 | -780374.21 | 154817.19 |
| | | | nf_gamma_10_1_100_True | 7.0 | -579285.19 | 156719.96 |
| | | 0.10 | nf_gamma_10_5_100_False | 8.0 | -484597.15 | 248508.84 |
| | | | nf_gamma_10_5_100_True | 10.0 | -277501.86 | 268802.25 |
| | | | nf_gaussian_0.1_0.001 | 5.0 | -421535.06 | 238781.26 |
| | | | nf_gamma_100_1_100_False | 10.0 | -286356.87 | 64751.89 |
| | | | nf_gamma_100_1_100_True | 10.0 | -48948.50 | 35427.55 |
| | | 1.00 | nf_gamma_100_5_100_False | 10.0 | -163008.27 | 158054.21 |
| | | | nf_gamma_100_5_100_True | 10.0 | -101276.96 | 77255.56 |
| | | | nf_gaussian_1_1e-2 | 8.0 | -825735.30 | 100242.24 |
| | | | nf_gamma_500_1_100_False | 10.0 | -161010.10 | 35334.52 |
| | | | nf_gamma_500_1_100_True | 10.0 | -140923.51 | 58184.19 |
| | | 5.00 | nf_gamma_500_5_100_False | 10.0 | -45679.41 | 21039.66 |
| | | | nf_gamma_500_5_100_True | 10.0 | -49196.44 | 56363.79 |
| | | | nf_gaussian_5_5e-2 | 10.0 | -147644.89 | 32270.53 |

Table 4: The model of interest is the realizable reduced rank regression model with $H$ hidden units and true weight $w_0 = (A_0, B_0)$ where $A_0$ and $B_0$ are as described in Section 7. The prior on the model parameters is taken to be $\varphi(w) = N(\mu(\varphi), \sigma^2(\varphi)I_d)$. For various combinations of values of $H$ and values for $(\mu(\varphi), \sigma^2(\varphi))$, we display the average and standard deviation of $\Psi(q, g)$ achieved after training on a dataset of size $n = 5000$ over 10 Monte Carlo iterations.

| $H$ | $(\mu(\varphi), \sigma^2(\varphi))$ | $\lambda_0/\beta_0 = \mu_0$ | method | count | mean | std |
|---|---|---|---|---|---|---|
| 24 | (5.0, 1.0) | 0.00 | nf_gaussian_0_1 | 10.0 | -87415.82 | 849.17 |
| | | | nf_gamma_10_1_100_False | 10.0 | -19679.77 | 43.94 |
| | | 0.10 | nf_gamma_10_1_100_True | 10.0 | -19633.08 | 59.80 |
| | | | nf_gamma_10_5_100_False | 2.0 | -20163.30 | 41.30 |
| | | | nf_gamma_10_5_100_True | 5.0 | -20066.54 | 100.67 |
| | | | nf_gaussian_0.1_0.001 | 10.0 | -19356.22 | 148.83 |
| | | | nf_gamma_100_1_100_False | 10.0 | -19936.36 | 42.41 |
| | | | nf_gamma_100_1_100_True | 10.0 | -19908.21 | 49.62 |
| | | 1.00 | nf_gamma_100_5_100_False | 6.0 | -20538.23 | 98.10 |
| | | | nf_gamma_100_5_100_True | 5.0 | -20237.83 | 146.99 |
| | | | nf_gaussian_1_1e-2 | 10.0 | -19974.33 | 49.01 |
| | | | nf_gamma_500_1_100_False | 10.0 | -20362.09 | 124.35 |
| | | | nf_gamma_500_1_100_True | 10.0 | -20367.47 | 122.89 |
| | | 5.00 | nf_gamma_500_5_100_False | 1.0 | -21345.31 | NaN |
| | | | nf_gamma_500_5_100_True | 1.0 | -21310.91 | NaN |
| | | | nf_gaussian_5_5e-2 | 10.0 | -24732.96 | 1798.18 |
| | (5.0, 100.0) | 0.00 | nf_gaussian_0_1 | 10.0 | -75338.44 | 767.59 |
| | | | nf_gamma_10_1_100_False | 10.0 | -8049.29 | 24.39 |
| | | 0.10 | nf_gamma_10_1_100_True | 10.0 | -8045.36 | 17.34 |
| | | | nf_gaussian_0.1_0.001 | 10.0 | -8012.99 | 13.94 |
| | | | nf_gamma_100_1_100_False | 10.0 | -8179.44 | 61.48 |
| | | | nf_gamma_100_1_100_True | 10.0 | -8110.42 | 44.60 |
| | | 1.00 | nf_gamma_100_5_100_False | 1.0 | -8675.67 | NaN |
| | | | nf_gamma_100_5_100_True | 1.0 | -8367.80 | NaN |
| | | | nf_gaussian_1_1e-2 | 10.0 | -8164.30 | 54.74 |
| | | | nf_gamma_500_1_100_False | 10.0 | -8472.94 | 107.69 |
| | | 5.00 | nf_gamma_500_1_100_True | 10.0 | -8510.36 | 107.44 |
| | | | nf_gaussian_5_5e-2 | 10.0 | -12594.11 | 1731.59 |
| 32 | (5.0, 1.0) | 0.00 | nf_gaussian_0_1 | 10.0 | -152649.27 | 934.00 |
| | | | nf_gamma_10_1_100_False | 10.0 | -34709.70 | 86.32 |
| | | 0.10 | nf_gamma_10_1_100_True | 10.0 | -34642.82 | 94.35 |
| | | | nf_gaussian_0.1_0.001 | 10.0 | -33988.99 | 181.45 |
| | | | nf_gamma_100_1_100_False | 10.0 | -35191.26 | 82.93 |
| | | 1.00 | nf_gamma_100_1_100_True | 10.0 | -35127.52 | 107.32 |
| | | | nf_gaussian_1_1e-2 | 10.0 | -35171.68 | 113.62 |
| | | | nf_gamma_500_1_100_False | 10.0 | -35978.89 | 225.51 |
| | | 5.00 | nf_gamma_500_1_100_True | 10.0 | -36019.50 | 239.84 |
| | | | nf_gaussian_5_5e-2 | 10.0 | -56902.20 | 13175.17 |
| | (5.0, 100.0) | 0.00 | nf_gaussian_0_1 | 10.0 | -131447.49 | 1093.16 |
| | | | nf_gamma_10_1_100_False | 10.0 | -14065.12 | 52.64 |
| | | 0.10 | nf_gamma_10_1_100_True | 10.0 | -14041.35 | 26.66 |
| | | | nf_gaussian_0.1_0.001 | 10.0 | -13990.93 | 31.67 |
| | | | nf_gamma_100_1_100_False | 10.0 | -14293.75 | 68.46 |
| | | 1.00 | nf_gamma_100_1_100_True | 10.0 | -14152.35 | 71.25 |
| | | | nf_gaussian_1_1e-2 | 10.0 | -14272.72 | 99.10 |
| | | | nf_gamma_500_1_100_False | 10.0 | -14874.08 | 126.36 |
| | | 5.00 | nf_gamma_500_1_100_True | 10.0 | -14906.96 | 144.99 |
| | | | nf_gaussian_5_5e-2 | 10.0 | -33611.96 | 10019.58 |

