# OpenReview forum: "Variational Inference via Resolution of Singularities"
_ICLR.cc/2022/Conference — ICLR 2022 Submitted_

### Official Review · Reviewer_32H4 · 2021-10-22

**Correctness:** 3
**Technical Novelty And Significance:** 2
**Empirical Novelty And Significance:** 2
**Recommendation:** 5
**Confidence:** 4

**Main Review:**

The writing is generally clear. Although there are innovations, such as using affine coupling network to learn the desingularization map, but I am not quite sure about the contribution of this paper. Equation (8) is misleading because it is not exact equality but should be $\asymp$, i.e., $\sup_{q\in\mathcal{Q}_{(0,1]}}\Psi_K(q,g)$ is bounded above and below by the right hand side up to constant multiples (see (15)). Therefore the lower bound of (8) is similar to the lower bound obtained by Bhattacharya et al. (2020) in (7). To improve upon their result, you need to show that $-\log{C(d)}$ is smaller than their constant.

I don't think recovering the leading order term of $\log{\bar{Z}_K(n)}$ guarantees that the mean field generalized gamma family is the optimal approximating family (as suggested in Pages 20-21 in Bhattacharya et al., 2020 and implicitly in this paper). There are many examples in the literature where dependent approximating families give clear improvement with respect to mean field (see "Variational Inference with Normalizing Flows" for example).

Here are some other questions:
1. Why do you work with $\Psi_K(q,g)$ and $\bar{Z}_K(n)$ in (4) and (5)? The results would be more significant if their stochastic counterparts $\Psi(q,g)$ and $\bar{Z}(n)$, which are the quantities of main interest, are considered instead.

2. Why did you choose the generalized gamma in constructing the mean field approximation? It seems to me that other distributions with tails heavier than Gaussian, such as the $t$-or Cauchy distributions, would also work.

#########################################################################################################
TYPOS:
1. In Section 2 the first display, there should be a $dw$ in $\bar{Z}(n)$
2. In Page 8 the last two paragraphs, write that Tables 1-4 are in Appendices D and E


**Summary Of The Paper:**

Working with Bayesian neural networks, this paper proposed a variational algorithm to approximate posterior distribution of the network weights. To overcome model singularities, the authors used the idea of normalizing flow by transforming the weights through an affine coupling network, and subsequently worked on the desingularized parameter space. In addition, they derived an asymptotic expression for the ELBO, and compared the Gaussian and generalized gamma approximating families in the experiments.

**Summary Of The Review:**

I have doubts about the contribution of this paper and its significance to the literature.

---

> ### Author Response · Authors · 2021-11-10
> **Response to Reviewer 32H4**
>
> *“Equation (8) is misleading because it is not exact equality but should be $\asymp$...Therefore the lower bound of (8) is similar to the lower bound obtained by Bhattacharya et al. (2020) in (7). To improve upon their result, you need to show that $-\log C(d)$ is smaller than their constant.”*
>
> Thanks for giving us the opportunity to clarify this. We decided to follow the convention in Lemma 3.1(iii) of Bhattacharaya which stated $\log B(k,h,\beta) \asymp -\lambda \log \beta$ and $G(\lambda, \beta) \asymp \lambda/\beta$. This is short hand for, respectively,
> $$
> \lim_{\beta \to \infty} \frac{|\log B(k,h,\beta) + \lambda \log \beta|}{\lambda \log \beta} =0
> $$
> and
> $$
> \lim_{\beta \to \infty} \frac{G(\lambda,\beta)}{\lambda/\beta} = 1
> $$
> We will correct Equation (8) accordingly with the $\asymp$ symbol as suggested.
>
> As for showing improvement on Bhattacharya's result, note that Theorem 3.1 in Bhattacharya first produces a lower bound on the quantity of interest, $\sup_q \Psi_K(q,g)$, and then shows the lower bound is $\asymp$ to $-\lambda \log n - constant$.
>
> *“I don't think recovering the leading order term of $\log \bar Z_K(n)$ guarantees that the mean field generalized gamma family is the optimal approximating family (as suggested in Pages 20-21 in Bhattacharya et al., 2020 and implicitly in this paper). There are many examples in the literature where dependent approximating families give clear improvement with respect to mean field (see "Variational Inference with Normalizing Flows" for example).”*
>
> There are two separate points raised in this comment. The first one is whether it is correct to say a variational family is optimal if it can recover the leading order term of $\log \bar Z_K(n)$? We do believe recovering the leading order term of $\log \bar Z_K(n)$ is desirable and perhaps “as good as it gets” since a gap between the ELBO and model evidence is always going to occur in reality. Of course, one could argue, why not try to match the higher order terms, wouldn’t that be even better? Possibly. But maybe matching the first order term is good enough for practice.
>
> The second point raised here is that non-mean-field families offer improvement over mean-field families. We agree! We did NOT use a mean-field family in $w$. The family $\mathcal Q$ in Equation (12) is mean field in $\xi$, not in $w$! Once the transformation $g$ is applied, the resulting variational approximation is NOT mean field in $w$, the neural network weight.
>
> Other questions:
> 1. We had acknowledged this as a limitation of the current work. As an initial line of inquiry, we feel it is acceptable to work with deterministic quantities. This was also done in Bhattacharya.
> 2. The generalised gamma source distribution was carefully reversed engineered to recover the leading order term of $\log \bar Z(n)$. We doubt the t or Cauchy distributions would yield the leading order term.
>
> Thanks for pointing out the typos. We will fix them.

---

> > ### Comment · Reviewer_32H4 · 2021-11-14
> > **Response to comments**
> >
> > Many thanks for your replies.
> >
> > 1. I believe that $\asymp$ means asymptotic equivalence, i.e., $a_n\asymp b_n$ means $C_1b_n\leq a_n\leq C_2b_n$ for constants $C_1\leq C_2$ (or $a_n=O(b_n)$ and $b_n=O(a_n)$) as $n\to\infty$. Point (iii) in Lemma 3.1 of Bhattacharya is not the definition for $\asymp$. Rather, $\log{B(k,h,\beta)}\asymp-\lambda\log{\beta}$ and $G(\lambda, \beta)\asymp\lambda/\beta$ follows from Point (iii). You can have a look at the paragraph before Section 1.2 in Page 4 of https://arxiv.org/pdf/1708.04753.pdf written by the same authors (Pati and Bhattacharya).
> >
> > 2. Your lower bound is $-\lambda\log{n}+\log{C(d)}$. However it is not clear to me whether $\log{C(d)}$ is positive or negative. If $\log{C(d)}>0$, then your lower bound is larger than Bhattacharya's and this is an improvement because your bounds are tighter. On the other hand, when $0<C(d)<1$, then $\log{C(d)}<0$ and your lower bound actually becomes $-\lambda\log{n}-|\log{C(d)}|$. When comparing to Bhattacharya's $-\lambda\log{n}-constant$, it is unclear whether your $|\log{C(d)}|$ is smaller or larger than $constant$. I think you need to show that $|\log{C(d)}|$ is smaller than $constant$ (for a tighter lower bound) in order for your result to be considered an improvement over Bhattacharya's.

---

> > > ### Author Response · Authors · 2021-11-16
> > > **about the \asymp notation and comparison to Bhattacharya**
> > >
> > > Thanks very much for your careful reading!
> > >
> > > 1. Thank you for pointing out our misunderstanding regarding the usage of $\asymp$.
> > > 2. If we take Theorem 3.1 in Bhattacharya and further assume that $b(\xi) \propto 1$ and $m=1$ (as we do in our Theorem 5.1), we can obtain  $\sup_{q \in \mathcal Q_[0,1]} \Psi_K(q,g) \asymp - \lambda \log n$ + constant free of $n$, in particular by setting $g$ to the resolution map, properly specifying all $\lambda_j, k_j$ in $q_j$, and setting $\beta_{j^*} =n$ and all other $\beta_j = 1$. (We had written minus constant in our Equation (7), but Bhattacharya uses plus constant so we'll correct this to be consistent.) Bhattacharya's constant is in fact given by the following expression
> > > $$
> > > A = \tilde \lambda  (1 - \prod_{j=m+1}^d G(\tilde \lambda_j, \beta_j)) + \sum_{j=m+1}^d [ \beta_j G(\tilde \lambda_j,\beta_j) + \log B(\tilde k_j, \tilde h_j, \beta_j) ]- \sum_{j=1}^d \log(2 \tilde k_j) - \sum_{j=1}^d \log(\tilde \lambda_j),
> > > $$
> > > evaluated at $\beta_j =1$ for $j\ge m+1$.
> > > The proof of our Theorem 5.1 shares the same initial steps as the proof of Bhattacharya’s Theorem 3.1 in that we also arrive at $-\tilde \lambda \log n + A$, see the equation immediately before (15). In the last steps, however, we argue that it is not wise to set $\beta_j = 1$ for all $j \ne j^*$ as done in Bhattacharya. Indeed, as long as $\tilde \lambda_j, j \ne j^*$ is large (which is true for DL), $A$ increases as $\beta_j$ increases.
> > >
> > > We can now see that we did not explain this properly in the manuscript. Also we can see that we neglected to disclose the caveat that $\tilde \lambda_j$ for $j \ne j^*$ needs to be large, which would be a property of the model-truth-prior triplet.

---

> > > > ### Comment · Reviewer_32H4 · 2021-11-16
> > > > **Response**
> > > >
> > > > Many thanks for the explanation!

---

> ### Author Response · Authors · 2021-11-10
> **Correction on the summary**
>
> *"To overcome model singularities, the authors used the idea of normalizing flow by transforming the weights through an affine coupling network, and subsequently worked on the desingularized parameter space.*"
>
> This requires a bit of clarification. Recall the resolution map links $\xi$ and $w$ via $g(\xi) = w$. You can think of $\xi$ as the desingularized parameter as you have written above. But note that we are, technically speaking, learning the transformation $g$ not $g^{-1}$. So it is more accurate to say we are using NFs to transform the desingularized weight $\xi$ through an affine coupling network.
>
> *"In addition, they derived an asymptotic expression for the ELBO, and compared the Gaussian and generalized gamma approximating families in the experiments."*
>
> Again, we want to clarify that we compared (generalized gamma source + affine coupling) to (std gaussian + same affine coupling network).  The approximating families in $w$ are not mean-field Gaussian nor mean-field generalized gamma.

---

> > ### Comment · Reviewer_32H4 · 2021-11-14
> > **Response to Correction on the summary**
> >
> > Thanks for your clarification!

---

### Official Review · Reviewer_jfja · 2021-10-28

**Correctness:** 3
**Technical Novelty And Significance:** 2
**Empirical Novelty And Significance:** 2
**Recommendation:** 5
**Confidence:** 3

**Main Review:**

Strength:
- The authors make an effort to exploit recent progress in singular model theory
for developing practical variational methods
- The summary of the related work is well presented, and appears to be comprehensive.
- The authors are clear about the limitations of their approach.

Weakness:
- Theorem 5.1. does not seem to tell more than Theorem 3.1 in [Bhattacharya et al. (2020)].
Am I missing here something?
- The main contribution seems to be the effort to exploit the theoretic idea of a mean-field variational approximation with truncated Gamma distributions. However, the relationship between the theoretic idea and their experiments appears ad-hoc since in their experiments they do not use a mean-field approximation, but rather use a normalizing flow with a Gaussian source distribution that
tries to approximate an untruncated Gamma distribution. Why is it an untruncated Gamma and not a truncated Gamma distribution?
What is the relationship between the mean-field approximation and the normalizing flow?
- Since the authors use a Gaussian approximation for the Gamma distribution, it would be interesting if the authors could show whether and when the approximation is justified in their setting. (It appears to be correct, since n is rather large)
- The presentation of the experimental results needs improvement.
The experimental results in Table 2ff are essential, but can only be found in the appendix.
They should be summarized in the main paper.
- Actually it is not clear to me why the Theorem 3.1 in [Bhattacharya et al. (2020)] suggests that
the ELBO found with a gamma mean-field approximation is better than a normal mean-field approximation.

**Summary Of The Paper:**

The paper builds on the recent theoretic results of [Bhattacharya et al. (2020)],
which shows that a mean-field variational approximation with carefully chosen approximation family
leads to an ELBO $\Psi$ which is sharp up to a constant C(d) which only depends on the dimensionality d of the parameter space.
For the proof, [Bhattacharya et al. (2020)] assumes that the approximation family is a Gamma distribution truncated to [0,1].
Therefore the authors of this paper conjecture that the Gamma distribution truncated to [0,1]
is a good choice for the source distribution of a normalizing flow.
Their experimental results suggest that this choice leads a higher ELBO than using as source distribution a normal distribution.

**Summary Of The Review:**

The idea of trying to use recent singular model theory to improve variational inference is interesting.
However, this work, in its current form, leaves too many open questions about the relation between theory and their experiments.
Furthermore, it is unclear what the take-home message is.

---

> ### Author Response · Authors · 2021-11-10
> **Response to Reviewer jfja**
>
> Response to summary:
>
> *"The paper...shows that a mean-field variational approximation with carefully chosen approximation family leads to an ELBO  which is sharp up to a constant C(d) which only depends on the dimensionality d of the parameter space."*
>
> We did *not* propose a mean-field approximation to the posterior distribution over the neural networks weights, $p(w|\mathcal D_n)$. Our approximation is non mean-field. Actually we also show that there exists $q$ and $g$ such that constant $C(d)$ can be recovered in Theorem 5.1.
>
> *"For the proof, [Bhattacharya et al. (2020)] assumes that the approximation family is a Gamma distribution truncated to [0,1]."*
>
> That is not an accurate characterization of Bhattacharya. The truncated gamma distribution is mean-field on $\xi$, not on $w$. Bhattacharya propose a non-mean-field family in $w$ actually.
>
> Response to weaknesses:
>
> *"Theorem 5.1. does not seem to tell more than Theorem 3.1 in [Bhattacharya et al. (2020)]. Am I missing here something?"*
>
> There is a lower bound in Theorem 3.1 in Bhattacharaya that makes it weaker than our Theorem 5.1
>
> *"The main contribution seems to be the effort to exploit the theoretic idea of a mean-field variational approximation with truncated Gamma distributions. However, the relationship between the theoretic idea and their experiments appears ad-hoc since in their experiments they do not use a mean-field approximation, but rather use a normalizing flow with a Gaussian source distribution that tries to approximate an untruncated Gamma distribution. Why is it an untruncated Gamma and not a truncated Gamma distribution? What is the relationship between the mean-field approximation and the normalizing flow?"*
>
> We did *not* use a mean-field approximation to $p(w|\mathcal D_n)$. The family $Q$ contains mean-field distributions in $\xi$. After the resolution map $g(\xi)=w$, the resulting distribution is not mean-field in $w$. You have misunderstood the basic premise. We did not use a normalizing flow with Gaussian source to approximate an untruncated Gamma as you state above. We use affine coupling layers to approximate the unknown resolution map $g$.
>
> At the end of Section 5 we state that because the assumption $b(\xi)$ is not critical, we consider the untruncated generalised gamma in the experiments.
>
>
> *"Since the authors use a Gaussian approximation for the Gamma distribution, it would be interesting if the authors could show whether and when the approximation is justified in their setting. (It appears to be correct, since n is rather large)*"
>
> This approximation has nothing to do with $n$. As we stated at the end of Section 6, if $V_j$ is gamma with large shape parameter $\lambda_j$ and rate $\beta_j$, then $V_j$ is approximately Gaussian. This approximation is quite elementary and non-controversial we think…? The approximation holds well for even moderately large $\lambda$.
>
> *"The presentation of the experimental results needs improvement. The experimental results in Table 2ff are essential, but can only be found in the appendix. They should be summarized in the main paper."*
>
> Thank you for the suggestion to summarize the Tables in the main text. (What is Table 2ff?) All tables are placed in the appendix due to space constraints.
>
> *"Actually it is not clear to me why the Theorem 3.1 in [Bhattacharya et al. (2020)] suggests that the ELBO found with a gamma mean-field approximation is better than a normal mean-field approximation.*"
>
> Bhattacharya makes no such claims. Again, Bhattacharya is not proposing gamma mean-field approximation. They are proposing gamma source + resolution map, and comparing it to mean-field Gaussian on $w$.
>
> We proposed gamma source + (affine coupling layers to approximation resolution map) and compared it to gaussian source + (same affine coupling layers).

---

> > ### Comment · Reviewer_jfja · 2021-11-20
> > **Thank you for the clarifications.**
> >
> > Thank you for the clarifications.
> > I now understand that the mean-field approximation with truncated gamma distributions is before the transformation $g(\xi)$, and therefore you want to use the truncated gamma distributions as a source distribution for the normalizing flow. You state in the paper that you "belief" that it is OK to replace the truncated by an untruncated gamma distribution. Can you give some intuition or evidence that supports your belief.

---

> > > ### Author Response · Authors · 2021-11-21
> > > **truncated to untruncated**
> > >
> > > That's right, in the paper we state that we believe it is OK to implement the untruncated gamma source distribution rather than the truncated version. This is because, as we also pointed out to Reviewer pvQE, we do not believe the assumption that $b(\xi) \propto 1$ is critical in Theorem 5.1. In particular when  $b(\xi)$ is not $\propto 1$, there will be a change to the formula (14) for the leading coefficient . The change will be a multiplicative factor that will not depend on $d$, so it is immaterial in that sense to the result in Theorem 5.1. (At the moment, we do not know exactly what the multiplicative factor is.) Now, when $b(\xi)$ is constant, we no longer have this constraint that $\xi$ reside in $[0,1]^d$.

---

> > > > ### Comment · Reviewer_jfja · 2021-11-23
> > > > **thank you**
> > > >
> > > > Thank you for the explanation.
> > > > I suggest adding your argumentation as a footnote or in the appendix.

---

### Official Review · Reviewer_PvQE · 2021-10-29

**Correctness:** 2
**Technical Novelty And Significance:** 2
**Empirical Novelty And Significance:** 2
**Recommendation:** 5
**Confidence:** 2

**Details Of Ethics Concerns:**

The ethics statement isn't really an ethics statement, but a limitation of the targeting higher ELBOs. Nevertheless, I don't have any ethics concerns.

**Main Review:**

Points marked with (*) are especially important, and should be addressed if the authors would like me to reconsider my score.

Strengths:
1) I enjoyed the discussing relating the ELBO score with the quality of the variational posterior predictive, and how good ELBO scores do not imply good posterior predictive distributions. Thank you for including this.

Weaknesses:
1) *As I understand, Eq. (1) shows for non-singular models, the posterior distribution can be expressed in a specific form i.e., a mixture of standard form distributions. However, this result only applies "when n is large". The discussion of what it means for "n to be large" is extremely limited. I expect this to depend on the network in question. Further, it is entirely unclear to me whether practical Bayesian Neural Networks are in a "large n" regime. This issue, at the very least, should be discussed within the paper.

2) *The technical contribution of the paper, at least in terms of a practical inference method, seems highly limited in terms of novelty. The paper proposes using a normalising flow approximate posterior with a Gamma source. However, for expressive flows, the choice of source distribution can always be absorbed into the flow. And flows have regularly been used for posterior inference. The contribution here is thus highly limited.
3) The readability and presentation of the paper could be improved. For example, equations could be named when referred to within the text. Intuition could be added on why neural networks are singular models. The figure captions are hard to understand without referring back to the text extensively. Experiments could be named, much more of the results could be in the main paper, intuition behind theorem conditions could be provided. I think this is especially valuable as singular learning theory is not yet well known within the machine learning community.
4) *The experimental evaluation of the paper is highly limited. I understand that simple networks are required to apply singular learning theory,  but the proposed method could be applied to much larger networks. The experiments within the related work, which is almost exclusively work within the Bayesian Deep Learning community, typically uses experiments with a much larger scale. Further, not all runs reached convergence, which undermines the claims made in the paper.
5) *The technical results of the paper assume that b \propto 1 (page 6), but the authors write that this condition is not expected to hold in reality, even for very simple experiments. This seems to be a major concern. This should, at the least, be mentioned when the theoretical results is discussed within the paper. The authors go on to write "we do not believe this assumption is critical", which as far as I can tell, is conjecture, and not convincingly discussed within the paper.
6) Although the authors use a Gamma source distribution, in practice, they approximate this with a Gaussian distribution. This seems to further undermine the technical novelty of the paper.
7) *The flow that is used within the paper is very shallow—it only has two coupling layers. It is well known that many layers of affine couplings to have expressive distributions. The authors claim "though learning the variational parameters, k, in the source distribution goes against conventional wisdom in normalizing flows, our experiments suggest some performance may be gained by learning the optimal untruncated generalized gamma source distribution at the same time as learning the resolution map". This claim is not well supported; the experimental results use particularly non-expressive flows, and as such, this claim only stands in this context. And within this context, I expect that learning parameters of the source distribution is increasing the expressivity of the flow. The conventional wisdom holds primarily for expressive flow distributions.
8) The authors write "for large n, the posterior is not Gaussian, but a mixture of standard forms". However, note that this is true only *post flow transformation. i.e., the distribution of the weights when transformed into a new coordinate system follows this form. However, in general, flows can express arbitrary distributions. This result seems to be somewhat oversold.

Clarifications:
1) As far as I can tell, the theory assumes that the space of parameters is a compact set. However, in BNNs, we typically place prior support over the entire reals. This seems to be a theory-practice mismatch. Is this problematic?
2) The asymptotic expansion of the normalised model evidence holds only for large values of n, right? I'm concerned about how this affects the validity of the theory.
3) The authors write "we are aided by the fact that a resolution map can attain the optimal value of Psi_K(q, g)". How/where is this demonstrated? Please clarify.
4) Singular learning theory provides this resolution map, which in practice is learnt using a normalising flow. However, the flow is actually learning by maximising the ELBO objective, which is a lower bound on the likelihood. I don't understand why **in the context of singular learning theory** which maximising the likelihood (approximately) is a sensible target for the flow. I think it is a sensible target, when considered in the context of variational inference.

**Summary Of The Paper:**

Inspired by asymptotic results from singular learning theory, the authors of the paper propose using a generalised gamma mean-field distribution with a normalising flow (that targets the "desingularization map") to perform variational inference. The authors additional build on prior work to derive a tighter bound on the log normalised evidence for variational inference using this approximate posterior combined with the "correct" desingularization map.

Note: while I am familiar with the Bayesian Neural Network literature, I am not familiar with singular learning theory. This is reflected in my confidence score. As such, I was unable to verify the theory in the paper, and assess the significance of the theoretic results included in the paper.

**Summary Of The Review:**

I have a number of concerns about the paper. In short, (i) limited technical novelty (ii) the theory does not seem to apply in practice (iii) the experimental evaluation is limited, and uses only small scale experiments (iv) room for improvement in terms of presentation e.g., many of the results are included only in the appendix.

As a result, I do not believe that this submission, in its current form, is of interest to the ICLR community.

---

> ### Author Response · Authors · 2021-11-09
> **Response to reviewer PvQE on weaknesses**
>
> Weaknesses
> 1. It does not seem too controversial to assume modern BNNs are in the large $n$ regime, since modern NNs are certainly deployed in the large $n$ regime?
> 2. We show that a particular source distribution pushed forward by a particular map (the resolution map) can achieve a small gap between the evidence and ELBO. Are there many normalizing flow papers out there that can characterize their approximation error? Perhaps the reviewer is uncomfortable with the novelty of how we implement our insight. Yes it would have been more novel to design a new architecture that exploits properties of a resolution map rather than the affine coupling layers we used in the experiments. This submission however is a good first step in highlighting that we ought to design networks that resemble resolution maps.
> 3. The equations are numbered and \eqref is used to reference them within the text. The text already references Murfet et al 2020 as to why NNs are singular. But we will include the example of permutation/scaling symmetries in ReLU networks making them non-identifiable with singular FIM as a simple way to understand why NNs are singular. The figure captions are brief due to space constraints. Thank you for the suggestion to name the experiments. We will add intuition for theorem conditions, but this will probably have to go in the Appendix due to space constraints.
> 4. Singular learning theory does not require simple networks. It applies to even deep neural networks. In our work, we only require simple networks insofar as we would like to know the RLCT $\lambda$, so that we can in turn know the term $-\lambda \log n$. (It is true that the RLCT is only known theoretically for simple networks.) It seems that without this baseline “truth”, we would be comparing the ELBOs of variational approximations that could all be very bad. But perhaps this is overcautious. As for the convergence issue, we believe this is due to not performing any type of hypertuning of learning rate, optimiser hyper parameters, etc.
> 5. When $b$ is not $\propto 1$, there will be a change to the formula (14) for the leading coefficient $C$. The change will be a multiplicative factor that will not depend on $d$, so it is immaterial in that sense. We will add this clarification.
> 6. Cannot see how this undermines novelty. We will readily admit aspects that are not novel in the work, for instance, using old-fashioned affine coupling layers to approximate the unknown resolution map. But the fact that the gamma distribution looks like the Gaussian distribution for certain parameter values has nothing to do with novelty.
> 7. Thank you for articulating this, we are inclined to agree with this observation. But again, it does not seem appropriate to frame this as a critique of the work. What if the source distribution does matter and the flow should mimic a resolution map, rather than relying on expressive flows and nondescript source distributions? What if properties of the resolution map allow us to use shallower flows? This paper suggests we should follow this line of inquiry.
> 8. There may be a misunderstanding here. The mixture of standard forms is a representation of the posterior distribution in the coordinate $\xi = g^{-1}(w)$. So that's pre-resolution map, not post.

---

> > ### Comment · Reviewer_PvQE · 2021-11-11
> > **Response to authors on weaknesses.**
> >
> > 1. What is a "large n" regime is not clear to me. For example, I think there remains underspecification in typical deep learning settings see e.g., [1], and I do not understand how this relates to "large n". The mixture of standard forms clearly occurs when we have posterior concentration (e.g., infinite data), and the mixture occurring due to symmetries in the weight space posterior. In other words, n must be large, but relative to what? What terms must shrink, and do we have credible evidence to believe that they are neglible in this case.
> >
> > 2. I don't believe that this paper characterises the approximation error of the practical implementation. Further, it is unclear whether n is sufficiently large for the mixture of normal form posterior to obtained in practice. In practice, the result here seems to be similar to the results for other flows: if the flow is sufficiently expressive, then there is zero gap. I don't find the evidence convincing that we ought to design flows that resemble resolution maps, due to the large n arguments (see above point).
> >
> > 3. Thank you.
> >
> > 4. Thank you; I understand that the theory applies to complex networks. But if the contribution is that "we should design flows that look like resolution maps", some evidence of this applying to deeper networks would make the argument stronger.
> >
> > 5. Thank you.
> >
> > 6. I believe that this undermines novelty because in practice, the approach used here looks closer to the "standard" flow setup.
> >
> > 7. The claim "our experiments suggest some performance may be gained by learning the optimal untruncated generalized gamma source distribution at the same time as learning the resolution map" is not well supported, since you use very inexpressive flows. Learning the source parameters increases expressivity, and perhaps it is simply this expressivity that increases performance. It could be interesting to see whether resolution map properties allow us to use shallower flows, but that is not what is studied in this paper. In general, it seems that changing the source distribution does not well "utilise" the insight of the paper.
> >
> > 8. Thank you.
> >
> > I have upgraded my score.
> >
> > [1] D'Amour, Alexander, et al. "Underspecification presents challenges for credibility in modern machine learning." arXiv preprint arXiv:2011.03395 (2020).

---

> > > ### Author Response · Authors · 2021-11-12
> > > **Response to PvQE on large $n$**
> > >
> > > Sorry about the continued confusion over "large $n$." The statement that for large $n$, the posterior distribution looks like a mixture of standard forms is based on the following rigorous statement from Singular Learning Theory:
> > > "Under natural conditions..., there exists a real analytic manifold $\mathcal M$ and a real analytic map $g: \mathcal M \to W$ such that
> > > $$
> > > K_n(g(u)) = u^{2k} - \frac{1}{\sqrt n} u^k \psi_n(u)
> > > $$
> > > where $\psi_n(u)$ converges in law to the Gaussian process $\psi(u)$. (This is Main Formula I in Watanabe's standard textbook "Algebraic Geometry and Statistical Learning Theory".)
> > >
> > > + I am saying "large $n$" in the same manner in which we might say the sample average $\bar X_n$ looks Gaussian when $n$ is large (central limit theorem) for a very broad class of distributions. If you're asking about the rate of convergence, it is $\sqrt n$, which is the classic convergence rate for many large-sample theorems in statistics.
> > >
> > > + "underspecification in typical deep learning settings". This is an interesting point! Throughout our paper, we assume that the truth is realizable by the model, i.e., there exists some $w_0$ such that $p_0(y|x) = p(y|x, w_0)$. We have already acknowledged this as a limitation of the work. There is quite a bit of work in SLT on the nonrealizable case. The theory still holds but is not as simple to state. That is why for this initial paper we work with the realizable assumption.
> > >
> > > + You also ask how is underspecification related to large $n$? Well, as explained just above, we assume there is no underspecification from the outset. This assumption is stated as early as the bottom of Page 1 in the submission.
> > >
> > > + "In other words, n must be large, but relative to what?" We are taking a classic $n \to \infty$ limit. We are *not* taking a random matrix theory type of limit where both $n$ and $d$ (number of parameters) go to infinity at some fixed ratio. (Although this would be very interesting.)
> > >
> > > + "What terms must shrink, and do we have credible evidence to believe that they are neglible in this case." Sorry I can't answer this at the moment because I don't understand what "terms" you are referring to.
> > >
> > > We're aware that the ICLR audience is not necessarily trained in classic statistics, so this seems partly the source of confusion on this large $n$ business. We are definitely happy to explain this better for the ICLR audience. Thanks for your comments on this matter.

---

> > > ### Author Response · Authors · 2021-11-12
> > > **Response to PvQE's response**
> > >
> > > 2a. *"I don't find the evidence convincing that we ought to design flows that resemble resolution maps, due to the large $n$ arguments."* For someone trained classically in statistics, large $n$ arguments form the very basis of many of our most celebrated theoretical and methodological results. But I suppose there's nothing we can do if the reviewer decides to reject large $n$ analyses. Finite $n$ analysis would be nice, but it is rarely ever tractable. Common behavior often emerges in the limit. Statisticians take advantage of this all the time, so do mathematical physicists. (Not to be too cheeky, but we're going to guess the reviewer accepts those universal approximation theorems of neural networks. Those are also asymptotic results.)
> > >
> > > 2b. *"I don't believe that this paper characterises the approximation error of the practical implementation."* It is true that the affine coupling layers we use may not be expressive enough to learn the resolution map. But in the same manner of speaking, NF flows in use may not be expressive enough to learn the posterior.
> > >
> > > 4. *"But if the contribution is that "we should design flows that look like resolution maps", some evidence of this applying to deeper networks would make the argument stronger."* We should be clear that there is 1) the neural network $p(y|x,w)$ that is used to model the relationship between output $y$ and $x$, and 2) there is another neural network that is used to learn the resolution map $g$. When you say we need evidence of application to deeper networks, we assume you mean when $p(y|x,w)$ is a deeper network? Theorem 5.1 absolutely holds when $p(y|x,w)$ is a deeper network. Maybe you are worried that none of our current experiments contain $p(y|x,w)$ that is a deep network. The two shallow $p(y|x,w)$ networks considered in the experiments were chosen because they have known RLCT $\lambda$. At the moment, RLCTs are not known for deep $p(y|x,w)$ networks. But we are now beginning to understand that the ICLR audience will not be satisfied that our experiments only consider shallow $p(y|x,w)$ networks. As we speak, we are repeating the experiments for deeper networks, where the true RLCT is not known.
> > >
> > > 6. We will have to respectfully disagree. Novelty, to us, doesn't mean that the end result has to be different, especially when there was little theoretical justification to accompany the first work to plant the flag.
> > >
> > > 7a. *"It could be interesting to see whether resolution map properties allow us to use shallower flows, but that is not what is studied in this paper."* Yes we agree that this is a conjecture for the moment. The classic NF approach to learning $p(w|\mathcal D_n)$ is to start with any source distribution + expressive flow to directly approximate $p(w|\mathcal D_n)$. But for singular models, this posterior is not nice looking! Our results in this paper says we can focus on designing an invertible NN to resemble the resolution map. The resolution map actually consists of a series of extremely simple operations, here's Example 44 from Watanabe's "matheatmical theory of Byesian statistics":
> > >
> > > The $K(w)$ function for this particular model-truth-prior triplet is given by $K(w_1,w_2) = \frac{1}{2\sigma^2} (w_1^4 - w_1^2 w_2 + w_2^3)^2$. A blow-up that resolves the singularities is given by
> > > $$
> > > w_1 = \xi_1, \quad w_2 = 3 \xi_1 \xi_2.
> > > $$
> > > This may seem like a particularly simple resolution map, but in fact in higher dimensions, the resolution map simply uses recursive blow ups. In conclusion, it may be easier to learn the nice-looking resolution map rather than the difficult-looking $p(w|\mathcal D_n)$. We will add this discussion to the paper.
> > >
> > >
> > > 7b. *"In general, it seems that changing the source distribution does not well "utilise" the insight of the paper."* Ah interesting! Please note that Theorem 5.1 states that there exists $q^* \in \mathcal Q$ with the desired property, which means that we are justified in learning the parameters of the source distribution.

---

> ### Author Response · Authors · 2021-11-10
> **Response to reviewer PvQE on clarifications**
>
> Clarifications
> 1. For most priors used, $\varphi(w)$ is practically zero outside of some compact neighborhood. We can then take the parameter space $W$ to be this compact neighborhood.
> 2. Yes the asymptotic expansion of the model evidence is for large $n$. Do you mean the validity of singular learning theory?
> 3. This is Theorem 5.1. The $g$ in Theorem 5.1 is a resolution map.
> 4. Thanks for giving us the opportunity to clarify this. Because there exists some $q^* \in \mathcal Q$ and a resolution map $g$ that maximizes $\Psi_K(q,g)$, we can look for $q^*$ and $g$ by maximizing the empirical version $\Psi(q,g)$.

---

> > ### Comment · Reviewer_PvQE · 2021-11-11
> > **Response to Reviewers on Clarifications**
> >
> > Thank you for these clarifications.
> >
> > 1. It seems that performing inference when restricting the parameter space is not the same as a prior with full support (e.g., in large data cases, perhaps inference would concentrate outside of the defined compact neighbourhood). I don't understand how this does not undermine the validity of the theory.
> > 2/3/4: thank you for clarifying.

---

> > > ### Author Response · Authors · 2021-11-11
> > > **compact neighborhood**
> > >
> > > We didn't mean to suggest that we should perform inference by restricting the parameter space. Rather the theory allows for a very large compact neighborhood outside of which the prior is machine epsilon zero.

---

### Official Review · Reviewer_kf3F · 2021-11-03

**Correctness:** 3
**Technical Novelty And Significance:** 2
**Empirical Novelty And Significance:** 2
**Recommendation:** 5
**Confidence:** 2

**Main Review:**

This is the first time I hear about the theory. I think the topic is very interesting.

However, the paper has several issues. First, I feel that the theoretical improvements over previous methods are not significant. Most theoretical results are just minor improvements such as making assumptions less restricted. The main result in 5.1 is still similar to [Bhattacharya et al. 2020], which already has the conclusion that a mean-field distribution allows the estimation of lambda and m.

Section 3 seems to be purely from previous work.

Second, the experiment is done on only two toy problems. The dataset has 5000 instances while the network has two hidden layers. Do you actually check whether the model is singular or not? I hope to see experiments based on realistic datasets. Can we also include other variational inference methods as the baseline when estimating the normalized evidence?

Third, I don't know how significant it is to use normalizing flow to learn the resolution map? Is the analysis still valid for the complex flow model? In practice, the method has competitors from other variational inference methods. How practice is this method in estimating the normalized evidence.

Fourth, the writing of the paper can be greatly improved. A lot of my understanding depends on previous work by [Bhattacharya et al. 2020]. Here is a list of detailed issues that block my understanding.
1. I think the paper should make clear the Laplacian approximation from the beginning and also the significance of estimating lambda and m.
2. Equation 6, "in the asymptotic expansion of ..." what expansion? The paper may need to explain the expansion a little.
3. Should section 3 go to the introduction since it is not the contribution of this paper?


**Summary Of The Paper:**

The paper analyzes the approximation of the log-ratio (called normalized evidence in the paper) (the log-probability under the model minus the log-probability under the true model). Since the model is singular, the Laplacian approximation of the normalized evidence needs a new method. The mean-field distribution and a resolution map allow us to estimate two important numbers (lambda and m) in the approximation.

The paper has made several contributions to the study of this problem in several aspects. First, the paper improves previous results by relaxing the assumption. Second, the paper actually uses a normalizing flow to actually learn the transformation.




**Summary Of The Review:**

The paper studies an interesting topic. However, there are several issues such as minor improvements over previous methods, practical value, and also writing problems. Overall, this is an interesting paper, but the quality is not very high.

---

> ### Author Response · Authors · 2021-11-09
> **Response**
>
> Reviewer kf3F may have seriously misunderstood certain essential aspects of the submission. The first sentence in the summary is incorrect. The log-ratio is not the normalised evidence. In no way do we analyse the approximation of the log-ratio. It is also not accurate to say “ the Laplacian approximation of the normalized evidence needs a new method.” The Laplace approximation does not need a new method. Rather the correct approximation of the normalised evidence is provided by singular learning theory. We never discuss in our work the estimation of $\lambda$ and $m$ (although it would be possible within the proposed framework).
>
> Yes Section 3 is an exposition on singular learning theory.
>
> NNs are singular because they are not identifiable and have singular FIM. We gave Murfet et al 2020 as a reference in the text. But we will clarify this point further in the manuscript.
>
> We did include NF with standard Gaussian source distribution as a baseline.
>
> 1. Not sure about this. We don't care about the Laplace approximation. It doesn't apply for singular models. Also, we are not trying to estimate $\lambda$ and $m$. Perhaps the reviewer only read Bhattacharya and not our submission?
> 2. We state it very plainly. The asymptotic expansion of $\bar Z_K(n)$.
> 3. No, we politely decline to move Section 3 on the grounds that it is not a “contribution.” We are not aware of a requirement to place all "non-contributions" in the Introduction.

---

> > ### Comment · Reviewer_kf3F · 2021-11-20
> > **Comment after more work**
> >
> > After checking other reviewers' comments, authors' replies, and many more hours on the original paper, I have more comments on the paper.
> >
> > 1. I have identified the following writing issues that make the paper hard to understand.
> > 1) The current version needs to have a better summary of the contribution. In particular, it needs to connect back the original problem of variational inference: tightening the gap. The authors' comment in "Significance and novelty" is a much better discussion than the current version of the paper.
> >
> > There is a contribution section at the end of section 2, but it is not explicit enough about its role in VI.
> >
> > 2) A lot of discussions are deep in the operations of symbols instead of the reasoning about the problem. An example here: "Finally, although we assume below that m = 1, this is not necessary; as long as m << d, we can set βj = n
> > 1/m for all j such that λ˜j = λ˜." The reader needs to go back to the previous section, check symbols, and then interpret the sentence. Another example, there is no explanation after equation (8). A sentence showing why it is significant for VI is very helpful.
> >
> > **Without a major revision, I don't think the paper is easy to understand for the general ICLR audience.**
> >
> > 2. I still think the contribution of the submission is limited. I don't find it making significant progress in VI for BNN. The authors seem to think that a resolution map is easier to approximate than the posterior -- "So perhaps, flows do not have to be very deep or expressive after all, it just has to look like a resolution map." -- but there is no support for such an argument.
> >
> > 3. The paper does not have large-scale experiments comparing the results with previous VI methods ([1], (Christos Louizos and Max Welling), and recent ones) with normalizing flows. Then it is hard to argue for the empirical value of the submission.
> >
> > In particular, the paper uses Gaussian distribution later in the empirical study. Though authors argue that "gamma distribution looks like the Gaussian distribution", but this may not be true in the high-dimensional case: note that q is a product of multiple distributions.
> >
> > 4. I also have two small questions about the development of the theory.
> > 1) The condition "Fundamental Conditions I and II with s = 2" of Theorem 3.1 should be discussed, so the applicable range of the theorem can be better understood.
> > 2) $M_{\alpha} = [0, b]^d$, does b depend on $\alpha$?
> >
> >
> > [1] Rezende, Danilo, and Shakir Mohamed. "Variational inference with normalizing flows." International conference on machine learning. PMLR, 2015.

---

> > > ### Author Response · Authors · 2021-11-21
> > > **thank you**
> > >
> > > *The current version needs to have a better summary of the contribution. In particular, it needs to connect back the original problem of variational inference: tightening the gap. The authors' comment in "Significance and novelty" is a much better discussion than the current version of the paper.*
> > >
> > > Thank you for the helpful feedback. We regret not having expressed our contribution better. We will incorporate our comment in "significance and novelty" as you suggested.
> > >
> > > *A lot of discussions are deep in the operations of symbols instead of the reasoning about the problem.*
> > >
> > > We'll take more care to offer insights in plain English.
> > >
> > > *there is no explanation after equation (8). A sentence showing why it is significant for VI is very helpful.*
> > >
> > > Good point!
> > >
> > > *In particular, the paper uses Gaussian distribution later in the empirical study. Though authors argue that "gamma distribution looks like the Gaussian distribution", but this may not be true in the high-dimensional case: note that q is a product of multiple distributions.*
> > >
> > > But the marginals are independent...
> > >
> > > *The condition "Fundamental Conditions I and II with s = 2" of Theorem 3.1 should be discussed*
> > >
> > > Good idea.
> > >
> > > *Does $b$ depend on $\alpha$?"
> > >
> > > No.
> > >
> > > *I still think the contribution of the submission is limited.*
> > >
> > > We've attempted to summarize our contribution better in our general comment titled "novelty and significance." We won't belabor the point.
> > >
> > > *"The authors seem to think that a resolution map is easier to approximate than the posterior -- "So perhaps, flows do not have to be very deep or expressive after all, it just has to look like a resolution map." -- but there is no support for such an argument.*
> > >
> > > This is a sentiment we expressed in the course of these discussions with the reviewers here. We realize we do not yet have rigorous proof. That's why we didn't write about this in the paper. Again, the contribution is simply to show a particular source distribution plus a resolution map can achieve the leading order term of the model evidence. That feels like (to us) a worthwhile theoretical contribution on its own.

---

### Author Response · Authors · 2021-11-09
**Significance and novelty**


We will be providing point-by-point responses shortly but would like to first take the opportunity to address the significance and novelty of our contribution. Let us say we agree that variational inference is an interesting tool for approximate Bayesian inference. Then we may ask ourselves, what constitutes a significant and novel contribution in variational inference. VI essentially contains two ingredients: 1) a variational family and 2) a divergence measure. We stick with the classic KL divergence for (2), so no innovation there. As for (1), the members of our variational family, at the end of the day, are generalized gamma source distributions pushed forward by affine coupling layers. Although this may resemble what you are familiarize with in the normalizing flow literature, the path to getting here is both unexpected and valuable.

We recognized that singular learning theory has something to say about the gap between the ELBO and the model evidence, particularly through the correct asymptotic expansion of the model evidence. So, we set out to use singular learning theory to derive a principled variational approximation to the posterior distribution over neural network weights.  Our analysis reveals that the generalized gamma source distribution pushed forward by a resolution map can achieve a reasonably small gap between the ELBO and the evidence. If we are not mistaken, there are no existing normalizing flow papers that theoretically address the gap between the ELBO and the evidence, in the context of variational approximation of the posterior distribution over neural network weights.

Instead, the current advances focus on ever more expressive flows. It's natural to think, sure there’s a gap, but if we have an expressive enough flow, we can forget about the gap. Also, not much attention is paid to the source distribution. This is understandable, as Reviewer PvQE points out, since the more expressive the flow is, the less the source distribution matters, perhaps.

This work however suggests another path. Specifically, the flow only has to look like a resolution map if we start with a generalized gamma source distribution. So perhaps, flows do not have to be very deep or expressive after all, it just has to look like a resolution map. Although the current work does not expend much effort to designing a flow that exploits the properties of being a resolution map, it is nonetheless a significant contribution to highlight that the flow ought to be targeting a resolution map.

---

### Author Response · Authors · 2021-11-09
**Confusion about mean-field**

We noticed that some reviewers might be under the wrong impression that we proposed a mean-field variational approximation for the posterior distribution over neural network weights $p(w|\mathcal D_n)$ in this work.

Note the family $\mathcal Q$ proposed in Equation (12) is mean-field in $\xi$. The variational distribution, a distribution in $w$, is induced by the transformation $g(\xi)=w$. The resulting variational distribution is NOT mean-field in $w$. This is why you see $(q,g)$ in many of the quantities in the paper, together they induce a variational distribution.

---

### Decision · Program_Chairs · 2022-01-20

**Decision:**

Reject

**Comment:**

The paper proposes a variational inference based on singular learning theory (SLT), where the resolution of singularity is learned by normalizing flow so that the latent distribution is factorized.

Pros:
- A unique idea to use SLT for variational inference.

Cons (only serious concerns):
- Goal is unclear.  The authors say that they propose variational inference based on SLT.  But apparently, they propose it not as an alternative to the state-of-the-art variational inference for neural networks (if so the experiments shown are far from the acceptable level).  The authors must clearly say for what purpose they propose a new method.  I would guess the proposed method is for analyzing singular models to compute their RLCT.  In that case, the authors should compare with existing methods for evaluating RLCT, e.g., MCMC based methods:

K. Nagata and S. Watanabe, "Exchange Monte Carlo Sampling From Bayesian Posterior for Singular Learning Machines," in IEEE Transactions on Neural Networks, vol. 19, no. 7, pp. 1253-1266, July 2008, doi: 10.1109/TNN.2008.2000202.

and discuss pros and cons of the proposed method.  For DNN, you should use the state-of-the-art MCMC sampling methods like

Wenzel, F., Roth, K., Veeling, B., Swiatkowski, J., Tran, L., Mandt, S., Snoek, J., Salimans, T., Jenatton, R. &amp; Nowozin, S.. (2020). How Good is the Bayes Posterior in Deep Neural Networks Really?. <i>Proceedings of the 37th International Conference on Machine Learning</i>, in <i>Proceedings of Machine Learning Research</i> 119:10248-10259 Available from https://proceedings.mlr.press/v119/wenzel20a.html.

as a baseline.  Approximating the posterior with normalizing flow can be another baseline.

- Large n issue.  SLT can be seen as a generalization of the asymptotic learning theory for the regular model, where the model complexity is represented by the parameter dimension d, and "asymptotic" means n >> d.  Watanabe revealed that the model complexity cannot be represented by d in singular models, and therefore the definition of "asymptotic" is not as clear as the regular case.  But it is known that typical neural networks are overparameterized and can achieve zero training error.  I have seen no work arguing that SLT holds in this regime.  If the authors insist that their method is applicable to deep neural networks, they should cite references where it would be proved that SLT holds in the overparameterized regime or prove it by themselves.

There are many more concerns including those pointed out by reviewers, and the paper is not ready for publication.